# Heritability and cross-species comparisons of human cortical functional organization asymmetry

Bin Wan[1,2,3,4]*, Şeyma Bayrak[1,3,4], Ting Xu[5], H Lina Schaare[1,4], Richard AI Bethlehem[6], Boris C Bernhardt[7], Sofie L Valk[1,4,8]*

[1]Otto Hahn Group Cognitive Neurogenetics, Max Planck Institute for Human Cognitive and Brain Sciences, Leipzig, Germany; [2]International Max Planck Research School on Neuroscience of Communication: Function, Structure, and Plasticity (IMPRS NeuroCom), Leipzig, Germany; [3]Department of Cognitive Neurology, University Hospital Leipzig and Faculty of Medicine, University of Leipzig, Leipzig, Germany; [4]Institute of Neuroscience and Medicine (INM-7: Brain and Behavior), Research Centre Jülich, Jülich, Germany; [5]Center for the Developing Brain, Child Mind Institute, New York, United States; [6]Department of Psychiatry, University of Cambridge, Cambridge, United Kingdom; [7]McConnell Brain Imaging Centre, Montréal Neurological Institute and Hospital, McGill University, Montréal, Canada; [8]Institute of Systems Neuroscience, Heinrich Heine University Düsseldorf, Düsseldorf, Germany

*For correspondence:
wanb.psych@outlook.com (BW);
valk@cbs.mpg.de (SLV)

Competing interest: The authors declare that no competing interests exist.

**Abstract** The human cerebral cortex is symmetrically organized along large-scale axes but also presents inter-hemispheric differences in structure and function. The quantified contralateral homologous difference, that is asymmetry, is a key feature of the human brain left-right axis supporting functional processes, such as language. Here, we assessed whether the asymmetry of cortical functional organization is heritable and phylogenetically conserved between humans and macaques. Our findings indicate asymmetric organization along an axis describing a functional trajectory from perceptual/action to abstract cognition. Whereas language network showed leftward asymmetric organization, frontoparietal network showed rightward asymmetric organization in humans. These asymmetries were heritable in humans and showed a similar spatial distribution with macaques, in the case of intra-hemispheric asymmetry of functional hierarchy. This suggests (phylo)genetic conservation. However, both language and frontoparietal networks showed a qualitatively larger asymmetry in humans relative to macaques. Overall, our findings suggest a genetic basis for asymmetry in intrinsic functional organization, linked to higher order cognitive functions uniquely developed in humans.

## Editor's evaluation

This is a valuable paper investigating hemispheric asymmetries in brain functional connectivity. The authors quantify this asymmetry using a solid methodology that capitalises on recent developments in functional gradients, and they further ask if these asymmetries are heritable and how they compare between humans and macaque monkeys. The results suggest a genetic underpinning of brain functional asymmetry, particularly in areas supporting unique human functions. These findings may help further our understanding of brain asymmetries.

## Introduction

The human cerebral cortex consists of two hemispheres that are not exactly alike and show marked differences in structure and function along a left-to-right axis (*Geschwind and Levitsky, 1968*; *Güntürkün et al., 2020*; *Karolis et al., 2019*; *Kong et al., 2018*; *Kong et al., 2022*; *Liang et al., 2021*; *Raemaekers et al., 2018*; *Sha et al., 2021*; *Zhong et al., 2021*). It has been suggested that the brain favors asymmetry to avoid duplication of neural circuitry having equivalent functions (*Karolis et al., 2019*; *Levy, 1977*). For example, bilateral cortical regions showing asymmetry in task-evoked activity have reduced (long-range) connections with the opposite homologous regions, favoring more local connectivity (*Karolis et al., 2019*).

Asymmetry, that is quantitative hemispheric differences between contralateral homologous regions, supports partly differentiable functional processes (*Karolis et al., 2019*; *Galaburda et al., 1990*; *Call et al., 2017*). Previous work has suggested that functions related to leftward dominance include language processing (*Lane et al., 2017*; *Piervincenzi et al., 2016*; *Wang et al., 2019*), letter search (*Pollack and Price, 2019*), and analogical reasoning (*Urbanski et al., 2016*). On the other hand, rightward dominance of functional activation has been related to holistic word processing (*Ventura et al., 2019*), visuospatial abilities (*Chen et al., 2019*), emotional processing (*Moeck et al., 2020*), as well as with psychiatric disorders such as autism spectrum disorder (*Floris et al., 2016*). In addition to task-related asymmetries, resting state functional connectivity (FC) studies have also reported hemispheric differences. For example, language areas of the middle and superior temporal cortex showed increased connectivity with regions in the left hemisphere relative to their right hemispheric counterparts (*Raemaekers et al., 2018*), and the right amygdala showed higher connectivity with the entire cortex than the left amygdala (*Tetereva et al., 2020*). Moreover, previous work has indicated that there are inter- and intra-hemispheric differences in functional connectivity between healthy adults and patients with schizophrenia (*Agcaoglu et al., 2018*), and between neurotypical individuals and those diagnosed with autism spectrum disorder (*Hahamy et al., 2015*). It is possible that such functional processing asymmetries may be driven by subtle differences in functional organization between the hemispheres.

One appealing approach to studying functional organization is by evaluating the low-dimensional axes, or gradients, present within the connectome. These approaches embed brain regions on a continuous data-driven space based on their functional connectome (*Coifman et al., 2005*; *Margulies et al., 2016*; *Vos de Wael et al., 2020*). Gradients capture how connectivity profiles from distinct cortical regions are integrated (i.e. similar functional connectivity profiles) and segregated (i.e. dissimilar functional connectivity profiles) across the cortex (*Margulies et al., 2016*; *Bethlehem et al., 2020*; *Haak and Beckmann, 2020*; *Paquola et al., 2019*). Regions that have similar connectivity profiles are at similar positions along these gradients, whereas regions with dissimilar connectivity profiles are placed further apart. The principal functional gradient, partly reflected in the intrinsic geometry of the cortex, shows that regions of the transmodal systems occupy locations equidistant from unimodal systems (*Margulies et al., 2016*; *Hong et al., 2020*; *Murphy et al., 2018*). Gradients provide a synoptic framework to capture smooth variations of connectivity patterns across the cortical mantle. They describe variations in genetic patterning (*Vainik et al., 2020*; *Valk et al., 2020*; *Valk et al., 2022*, functional processes *Margulies et al., 2016*; *Murphy et al., 2018*; *Turnbull et al., 2020*), and are observed across species (*Valk et al., 2020*; *Coletta et al., 2020*; *Xu et al., 2020*). Gradients have been linked to graph-theoretical markers such as degree centrality (*Hong et al., 2019*) and microcircuit dynamics (*Park et al., 2021*) as well as connectivity distance (*Hong et al., 2019*; *Wang et al., 2021*). Moreover, the principal gradient describes the geodesic distance between primary and default regions, and relates to cortical microstructure and associated processing hierarchies (*Huntenburg et al., 2018*). In doing so, and in contrast to clustering or network-based approaches, the gradient framework provides a spatial ordering of functional brain networks, placing them along a gradual axis of connectivity variation reaching from sensory to transmodal areas. In the context of asymmetry of gradient loadings this would mean that a given region with a significant left-ward asymmetry along the first gradient (sensory-to-transmodal) has a connectivity profile more similar to the transmodal anchor in the left hemisphere relative to the right. Consequently, these regions are placed at different positions along the cortical hierarchy, providing novel insights concerning the system-level variations in the asymmetric brain. Indeed, recent research suggests that the principal gradient is asymmetric (*Liang et al., 2021*; *Gonzalez Alam et al., 2022*) and that the degree of

asymmetry relates to individual differences in semantic performance and visual reasoning (*Gonzalez Alam et al., 2022*).

As inter-hemispheric asymmetry has been observed consistently in human brain structure and function, there may be important (phylo)genetic factors supporting lateralized human cognition (*Güntürkün et al., 2020*; *Corballis, 1989*; *Corballis, 2009*; *Güntürkün and Ocklenburg, 2017*; *Vilain et al., 2011*; *Vallortigara et al., 1999*). Previous work has reported that brain structure asymmetry is heritable (*Sha et al., 2021*; *Zhong et al., 2021*), especially in the language areas, and differentiates between humans and non-human primates (*Eichert et al., 2020*; *Eichert et al., 2019*; *Neubauer et al., 2020*; *Spocter et al., 2020*). At the same time, it has been shown that both humans and apes show asymmetry of brain shape (*Neubauer et al., 2020*), indicating that asymmetry is not a uniquely human brain feature. However, asymmetry was observed to be more local and variable in humans, potentially suggesting that individual variation in asymmetry in humans varies as a function of localized networks rather than global features. It is of note that the full FC matrix contains both intra-hemispheric and inter-hemispheric connections. Intra-hemispheric connections, compared to the inter-hemispheric connections, have been suggested to reflect inhibition of the corpus callosum and may underlie hemispheric specializations involving language, reasoning, and attention (*Gazzaniga, 2000*). Conversely, inter-hemispheric connectivity may reflect information transfer between hemispheres, for example, of motoric information, or crude information concerning spatial locations (*Gazzaniga, 2000*). Previous studies have mainly employed intra-hemispheric FC to study gradient asymmetry (*Liang et al., 2021*; *Gonzalez Alam et al., 2022*). However, inter-hemispheric differences in functional connectivity may also have functional relevance. For example, inter-hemispheric connectivity has been reported to be abnormal in patients with schizophrenia (*Agcaoglu et al., 2018*; *Chang et al., 2019*) and autism (*Hahamy et al., 2015*). Indeed differences not only of functional organization within each hemisphere but also between hemispheres, enabled by the corpus callosum, are relevant for integration and segregation of cognitive function and support hemispheric coordination (*Gazzaniga, 2000*; *Toga and Thompson, 2003*).

Here, we investigated the genetic basis of asymmetry of functional organization. We first examined whether inter-individual differences in asymmetry of functional organization are under genetic control, that is heritable. Second, we investigated whether asymmetry of functional organization is phylogenetically conserved in macaques. To probe individual variation in asymmetry of functional organization, we utilized a data-driven nonlinear dimension reduction technique, as this approach can provide reliable and robust indices of individual variation of cortical organization (*Hong et al., 2020*). We first obtained connectomic gradients for each hemisphere separately (left and right intra-hemispheric) as well as those describing functional connectivity from left to right and right to left hemispheres (left and right inter-hemispheric). We then computed the difference between individual gradient scores to study the asymmetry, consistent with prior studies (*Liang et al., 2021*; *Gonzalez Alam et al., 2022*). Subsequently, to evaluate the heritability of possible differences between left and right intra- and inter-hemispheric FC gradients, we used the twin pedigree set-up of the Human Connectome Project S1200 release young adults dataset (*Van Essen et al., 2013*). To assess whether asymmetry is conserved in other primates, we compared the asymmetry of functional gradients of humans with those observed in macaque monkeys using the prime-DE dataset (*Xu et al., 2020*; *Milham et al., 2018*). Finally, we conducted a confirmatory meta-analysis to explore the relationship between the patterns of gradient asymmetry and task-based functional MRI activations. Multiple analyses verified the robustness and replicability of our results.

## Results

### Hemispheric functional connectivity gradients (Figure 1)

To obtain intra-hemispheric gradients, we first computed the functional connectivity (FC) in 180 homologous parcels per hemisphere using a multimodal parcellation (MMP, *Glasser et al., 2016*) for each subject (n=1014). For the network level analyses, we employed the Cole-Anticevic atlas (*Ji et al., 2019*) based on the MMP (*Figure 1a*). For each individual, FC was summarized in two different patterns (*Figure 1b*): FC within the left hemisphere (LL mode, intra-hemispheric pattern), within the right hemisphere (RR mode, intra-hemispheric pattern), from left to right hemisphere (LR mode, inter-hemispheric pattern), and from right to left hemisphere (RL mode, inter-hemispheric pattern). We

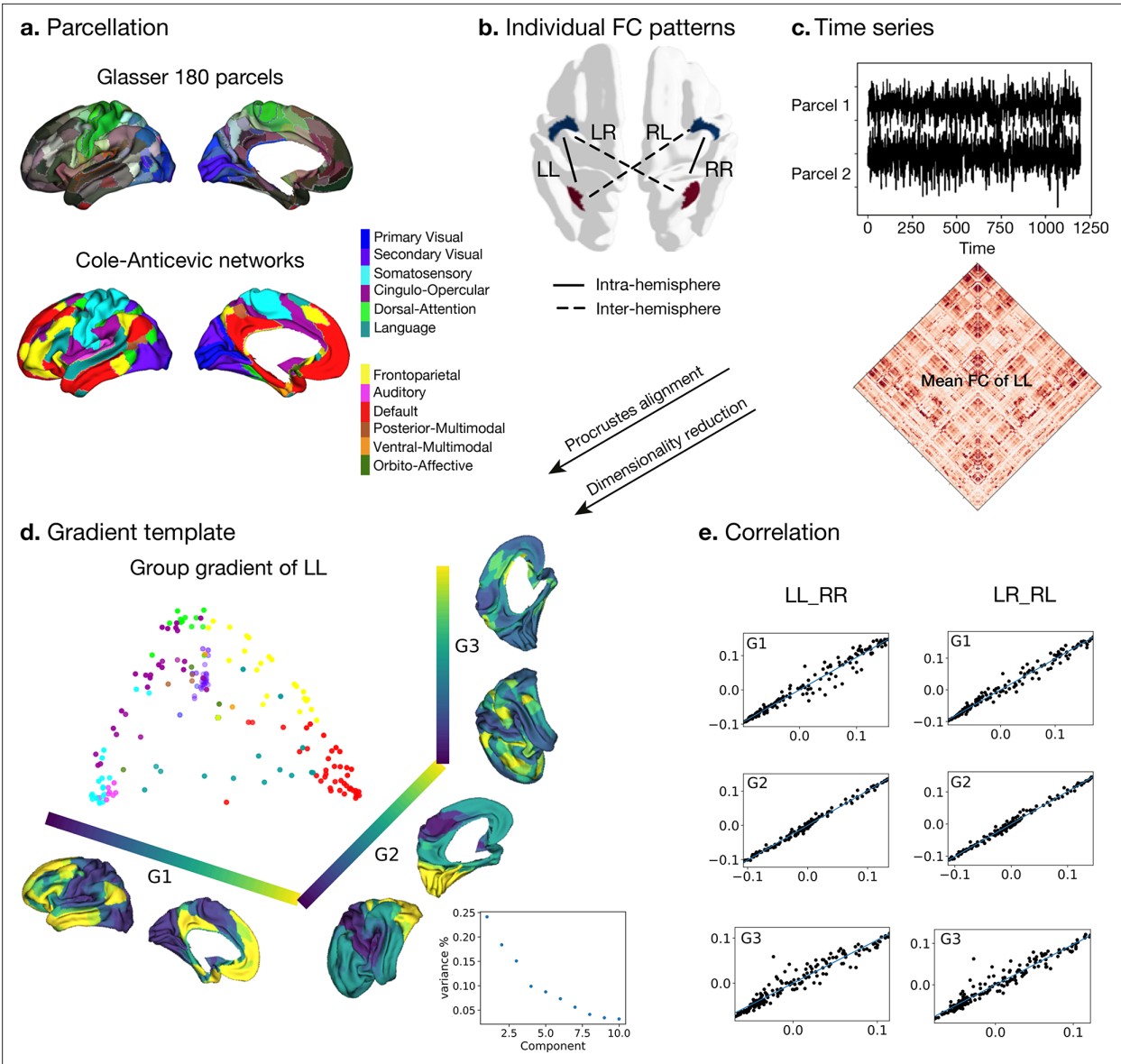

**Figure 1.** Processing of functional gradients in humans. (**a**) Parcellation using Glasser atlas (**Glasser et al., 2016**) in each hemisphere and Cole-Anticevic (CA) networks (**Ji et al., 2019**) for humans. (**b**) Individual FC in each hemispheric pattern, that is left-left (LL, intra-hemisphere), right-right (RR, intra-hemisphere), left-right (LR, inter-hemisphere), and right-left (RL, inter-hemisphere). (**c**) Time series of two parcels and the mean functional connectivity (FC) matrix between left and left hemisphere (LL). (**d**) Gradient template using the group-level gradient of LL. Dots represent parcels and are colored according to CA networks. The decomposition scatter on the right below depicts x-axis (number of eigenvectors) and y-axis (the contribution of each eigenvector to the total). (**e**) Correlation between left and right mean gradients across subjects of intra- and inter-hemispheric patterns. Left panel is the correlation between gradients of FC LL and FC RR (intra-hemispheric pattern). Right panel is the correlation between gradients of FC LR and FC RL (inter-hemispheric pattern). All correlation coefficients along G1, G2, and G3 are greater than 0.9.

The online version of this article includes the following figure supplement(s) for figure 1:

**Figure supplement 1.** Mean asymmetry index along G4-10 across subjects in humans.

**Figure supplement 2.** Individual gradients of each FC pattern.

selected the LL mode as the reference template for the gradients approach, and therefore assessed the mean FC that was determined by averaging LL FC across subjects (lower panel in *Figure 1c*). Here, the reference matches the order and direction of the gradient but does not rescale the gradients. The template gradients were computed by implementing diffusion map embedding, a non-linear dimension reduction technique (*Coifman et al., 2005*), on the mean LL FC using BrainSpace (*Vos de Wael et al., 2020*). The current study analyzed asymmetry and its heritability using the first three gradients

that explained the most variance (*Figure 1d*). Each gradient has reasonably well-described functional associations (G1: unimodal-transmodal gradient with 24.1%, G2: somatosensory-visual gradient with 18.4%, G3: multi-demand gradient with 15.1%). However, given that we extracted 10 gradients to maximize the degree of fit (*Margulies et al., 2016*; *Mckeown et al., 2020*). We describe mean asymmetry of G4-10 in *Figure 1—figure supplement 1*.

Next, individual gradients were computed for each subject and the four different FC modes and aligned to the template gradients with Procrustes rotation. It was applied without a scaling factor so that the reference template only matters for matching the order and direction of the gradients. The procedure rotates a matrix to maximum similarity with a target matrix minimizing the sum of squared differences. As noted, Procrustes matching was applied without a scaling factor so that only the reference template matters for matching the order and direction of the gradients. Therefore, it allows comparison between individuals and hemispheres. The individual mean gradients showed high correlation with the group gradients LL (all Spearman $r>0.97$, $P_{spin} <0.001$). *Figure 1e* shows the correlation between LL and RR, LR, and RL modes. In each case, the gradients were highly similar. Similar to previous work (*Coifman et al., 2005*) we observed that the principal gradient (G1) traversed between unimodal regions and transmodal regions (e.g. default-mode network: DMN) whereas a visual to somatosensory gradient was found for G2. The tertiary gradient (G3) dissociated control from DMN and sensory-motor networks (*Figure 1d and e*, and *Figure 1—figure supplement 2*). We employed spin permutations for correcting spatial Spearman correlation p values, that is *p spin*. For the intra-hemispheric pattern, the mean gradients of LL were strongly correlated with those of RR (Spearman $r_{G1}=0.988$, $P_{spin} <0.001$, $r_{G2}=0.989$, $P_{spin} <0.001$, $r_{G3}=0.967$, $P_{spin} <0.001$). For the inter-hemispheric pattern, the mean gradients of LR were also strongly correlated with those of RL (Spearman $r_{G1}=0.993$, $P_{spin} <0.001$, $r_{G2}=0.985$, $P_{spin} <0.001$, $r_{G3}=0.969$, $P_{spin} <0.001$).

## Asymmetry of functional gradients in humans (Figure 2)

Next, we computed the asymmetry index (AI) by subtracting the right hemispheric gradient scores of each parcel from the corresponding left hemispheric scores for our intra- and inter-hemispheric connectivity patterns (*Figure 2a*). A red AI indicates rightward dominance in gradient scores, whereas blue indicates leftward dominance. The differences in gradient loadings (parcel No.25: Peri-Sylvian language area) reflect differences in connectivity profiles (top 10%) between LL versus RR, or LR versus RL, respectively (*Figure 2—figure supplement 1*). The significance of AI scores for the intra- and inter-hemispheric patterns were reported after false discovery rate adjustment ($P_{FDR} < 0.05$) (*Figure 2b*), and Cohen's d maps can be seen in *Figure 2—figure supplement 2*. Frontal and temporal lobes showed the greatest intra-hemispheric asymmetry in G1 (*Supplementary file 1*). In particular, regions in ventral- and dorsolateral PFC (11 l, p9-46v, p10p) were the three most rightward asymmetric areas and regions in temporal polar cortex, dorso/posterior superior temporal sulcus, and inferior frontal gyrus (TGv, STSdp, and 55b) were the three most leftward asymmetric areas in the intra-hemispheric pattern. Network-level analyses (*Figure 2c*) indicated that the language ($t=41.3$, df = 1013, $P_{FDR} < 0.001$) and default mode ($t=17.3$, df = 1013, $P_{FDR} < 0.001$) networks had a high leftward AI, while the frontoparietal network ($t=-26.0$, df = 1013, $P_{FDR} < 0.001$) had a high rightward AI. We observed no significant difference of AI in primary and secondary visual networks. Overall, asymmetry was widely present along the first three connectivity gradients, including G2 and G3. Detailed numbers can be seen at online ipython notebook (code availability).

For the inter-hemispheric pattern, a large portion of the cerebral cortex showed significant AI scores. The top six asymmetric areas included regions in inferior frontal cortex and parahippocampal regions (11 l, 47 m, p9-46v, PSL, PreS, and PHA2) (*Supplementary file 1*). At the network level (*Figure 2c*), networks with leftward dominance were the visual ($t_{primary\ visual} = 9.3$, df = 1013, $P_{FDR} < 0.001$; $t_{secondary\ visual} = 7.5$, df = 1013, $P_{FDR} < 0.001$), language ($t=5.7$, df = 1013, $P_{FDR} < 0.001$), default mode ($t=11.9$, df = 1013, $P_{FDR} < 0.001$), and orbito-affective ($t=4.6$, df = 1013, $P_{FDR} < 0.001$) networks. Networks with rightward dominance were the somatomotor ($t=-3.5$, df = 1013, $P_{FDR} = 0.00059$), cingulo-opercular ($t=-14.6$, df = 1013, $P_{FDR} < 0.001$), dorsal attention ($t=-8.0$, df = 1013, $P_{FDR} < 0.001$), frontoparietal ($t=-12.1$, df = 1013, $P_{FDR} < 0.001$), and auditory ($t=5.7$, df = 1013, $P_{FDR} < 0.001$) networks. Posterior and ventral multimodal networks were not significantly asymmetric.

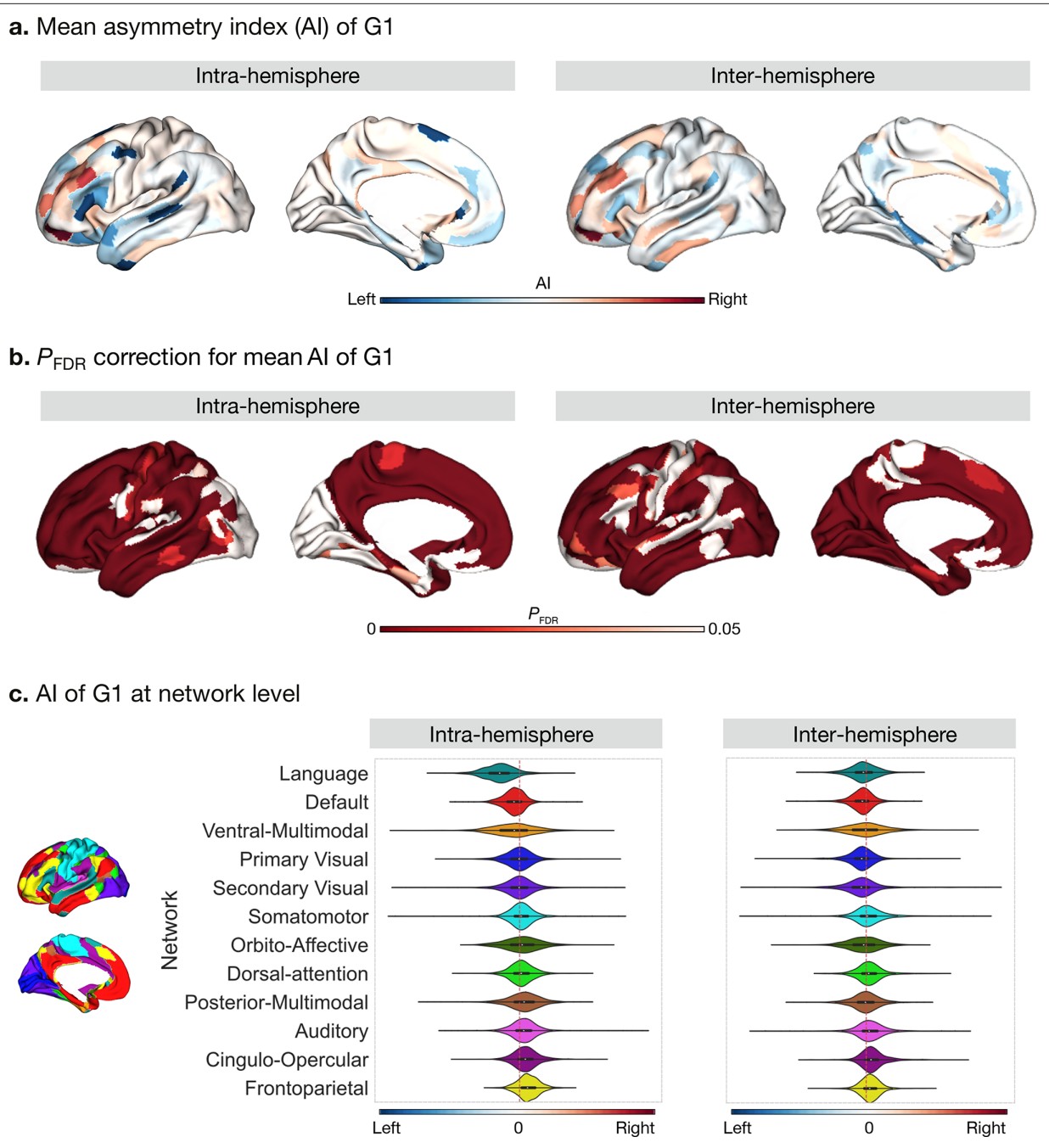

**Figure 2.** Asymmetry of functional gradients in humans and its heritability. (**a**) Mean asymmetry index (AI) of intra- and inter-hemispheric patterns in humans. Red and blue indicate rightward and leftward asymmetry respectively. (**b**) FDR correction for the P values of AI shown in A; (**c**) Violin plots of mean AI network loading across individuals (n=1014), with median, 25%-75%, and distribution at 25/75% -/+1.5 interquartile range. Networks are ranked from leftward (language) to rightward asymmetry (frontoparietal) along the intra-hemispheric principal gradient.

The online version of this article includes the following figure supplement(s) for figure 2:

**Figure supplement 1.** FC profiles of the most asymmetric parcel (No.25: Peri-Sylvian language area).

**Figure supplement 2.** Cohen's d maps of G1 asymmetry in humans.

**Figure supplement 3.** Cohen's d (yellow-purple) and PFDR (red) of asymmetry index with (left-right)/(left +right).

**Figure supplement 4.** Asymmetry using Desikan-Killiany (DK) atlas.

**Figure supplement 5.** Asymmetry using UK Biobank sample (n=34,830).

**Figure supplement 6.** Vertex-wise asymmetry along G1.

*Figure 2 continued on next page*

**Figure supplement 7.** Asymmetry of functional gradients using HCP.

**Figure supplement 8.** Asymmetry of functional gradients using UKB.

**Figure supplement 10.** Correlations between LL aligned gradients asymmetry and RR aligned gradients asymmetry.

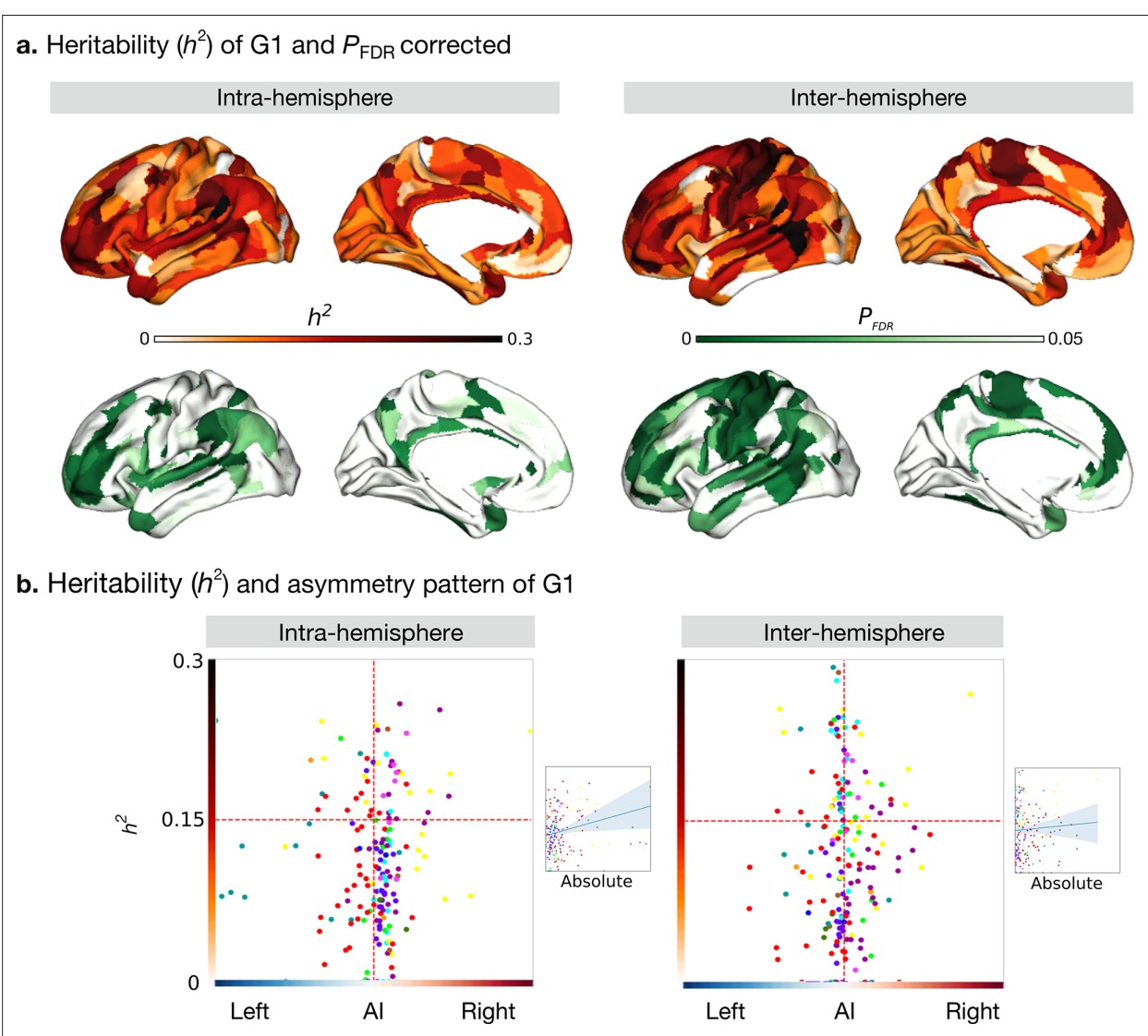

**Figure 3.** Heritability of asymmetry of functional G1. (**a**) Heritability (orange colorbar) and p values after FDR correction (green colorbar). (**b**) Scatter plot of heritability and AI scores. The x- and y-axes are the mean asymmetry index and heritability, respectively. Dots represent parcels and are colored according to CA networks. The small scatter plots with a regression line are the corresponding absolute mean asymmetry index (x-axis) and heritability (y-axis).

The online version of this article includes the following figure supplement(s) for figure 3:

**Figure supplement 1.** Mean asymmetry index (AI), heritability (h2), and PFDR of G2 and G3.

The mean AI scores across individuals for the intra- and inter-hemispheric patterns showed high similarity (Spearman $r_{G1}$ = 0.645, $P_{spin}$ <0.001). This may indicate that the asymmetric functional organization is a feature that is captured both by inter- and intra-hemispheric connectivity patterns.

## Heritability of asymmetry of functional gradients in humans (Figure 3)

We next computed the heritability of the AI scores of the functional gradient for the intra- and inter-hemispheric patterns using Solar-Eclipse 8.5.1 beta (http://solar-eclipse-genetics.org/). We found that left-right differences observed in large-scale functional organization axes were heritable (*Figure 3a*). Specifically, for the intra-hemispheric pattern, we found sensory-motor regions, middle temporal regions, dorso-lateral, and medial prefrontal regions to be heritable ($P_{FDR}$ < 0.05). In the case of the inter-hemispheric pattern, all cortical regions with the exception of visual areas and superior temporal and insular regions were heritable ($P_{FDR}$ < 0.05). Notably, language-associated areas such as the PSL (Peri-Sylvian language area) and 55b had the highest heritability in both the hemispheric patterns (PSL: intra: $h^2$=0.46, $P_{FDR}$ < 0.001 and inter: $h^2$=0.34, $P_{FDR}$ < 0.001, *Supplementary file 1*). However, BA area 44 (Broca's area) showed low heritability (intra: $h$*Güntürkün et al., 2020* = 0.12, $P_{FDR}$ = 0.026 and inter: $h^2$=0.12, $P_{FDR}$ = 0.018). The G2 and G3 results are shown in *Figure 3—figure supplement 1*.

To assess whether regions showing higher asymmetry had an increased heritability of G1, we plotted our cortical maps of asymmetry along those reporting heritability (*Figure 3b*). For the correlation between the absolute asymmetry index and heritability (*Figure 3b* small scatter), gradients of the intra-hemispheric FC patterns were significant (Pearson $r$=0.245, $P_{spin}$ = 0.005) while gradients of the inter-hemispheric FC were not (Pearson $r$=0.055, $P_{spin}$ = 0.613).

## Asymmetry of functional gradients in macaques (Figure 4)

To probe the phylogenetic conservation of asymmetry of functional organization in primates, we performed the same diffusion map embedding analysis on macaque resting-state FC data (n=19, PRIMATE-DE sample *Xu et al., 2020*; *Milham et al., 2018*). We used the Markov parcellation (*Markov et al., 2014*) in macaques, resulting in 91 parcels per hemisphere (*Figure 4a*) and then computed FC in the four patterns: LL and RR (intra-hemispheric patterns), and LR and RL (inter-hemispheric patterns). Following the same connectome gradients analysis pipeline as deployed on the human FC data, we obtained the template gradients on the LL intra-hemispheric FC pattern (*Figure 4b*). The first three template gradients explained 20.0%, 15.2%, and 12.8% of total variance, respectively. G1 described an axis traversing dorsolateral prefrontal and parietal regions (anterior-posterior).

Evaluating the intra-hemispheric pattern of functional organization in macaques along G1, we observed that parietal cortices had a rightward dominance while occipital cortices were leftward. Temporal cortex asymmetry was low (*Figure 4c*). The inter-hemispheric pattern showed similar asymmetry to the intra-hemispheric pattern along G1. However, the AI scores of the principal, but also secondary and tertiary gradients, were not statistically significant after FDR correction, both for intra- and inter-hemispheric patterns. The effect sizes across cortex observed in macaques along G1 were [intra: mean Cohen's d=–0.27 (rightward) and 0.27 (leftward); inter: mean Cohen's d=–0.22 (rightward) and 0.20 (leftward)].

To compare human and macaque connectomic gradients, we aligned human gradients to the same macaque surface space (*Figure 4d*) using a joint embedding technique (*Xu et al., 2020*). We summarized Cohens' d of AI of macaque-aligned human gradients within the Markov parcels for the intra- and inter-hemispheric patterns and compared the similarity of Cohens' d of AI between the two species using Spearman correlations (*Figure 4e*). To reduce the systematic bias during the cross-species alignment, we averaged the results of left and right hemispheric alignment. We found that the macaque and macaque-aligned human AI maps of G1 were correlated positively for intra-hemispheric patterns (Pearson $r$=0.345, $P_{spin}$ = 0.030). For inter-hemispheric patterns, we did not observe a significant association (Pearson $r$=–0.029, $P_{spin}$ = 0.858).

We then projected the human functional networks (*Ji et al., 2019*) on the macaque surface (*Xu et al., 2020*), to qualitatively compare differences in human functional networks between humans and macaques (*Figure 4f* and *Figure 4—figure supplement 1*). In the case of the intra-hemispheric asymmetry of the principal FC gradient, we observed that humans showed high leftward asymmetry in the language and default mode networks but macaques did not. Moreover, humans showed high rightward asymmetry in the frontoparietal and cingulo-opercular networks but macaques did not.

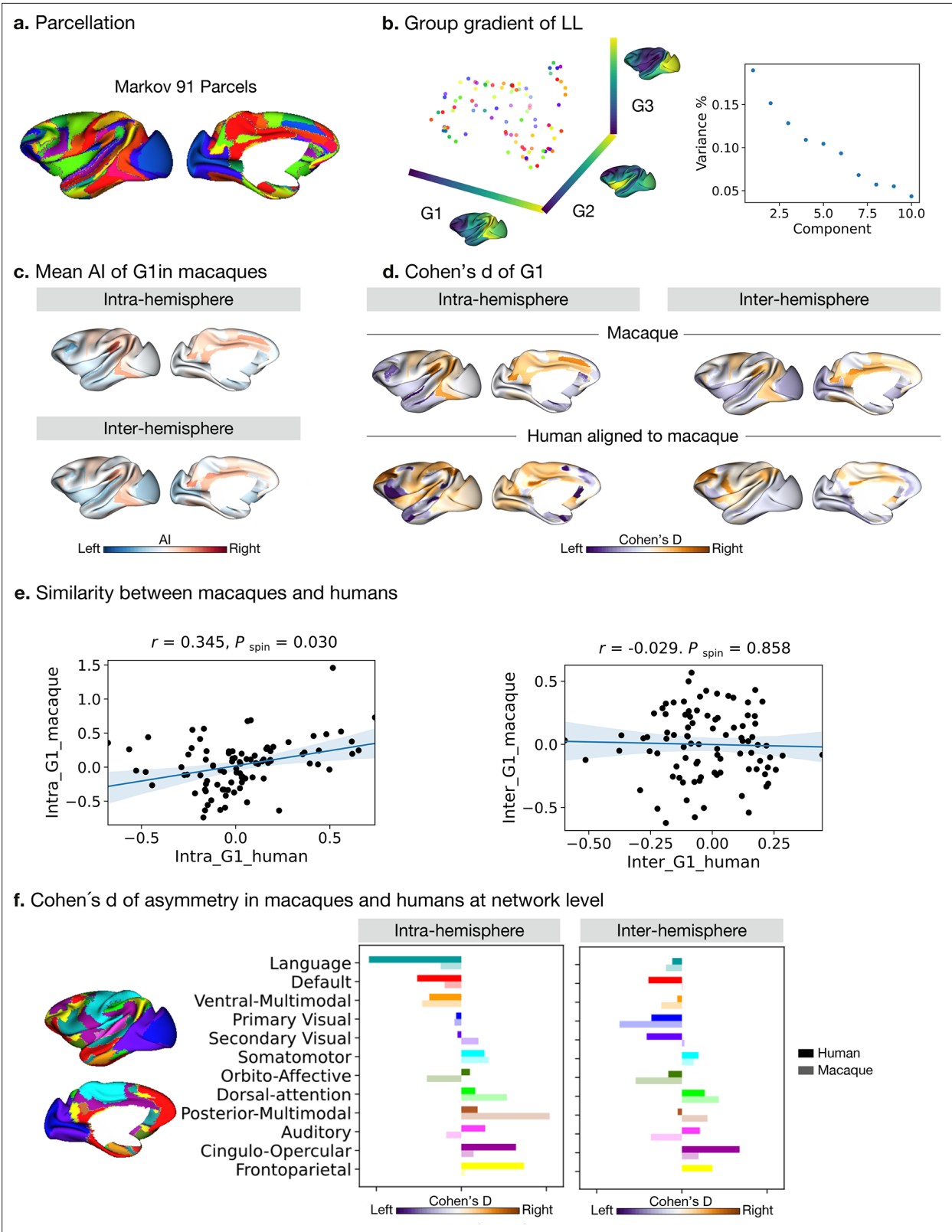

**Figure 4.** Asymmetry of functional gradients in macaques. (**a**) Parcellation used Markov atlas in macaques *Markov et al., 2014*. (**b**) Template gradients of group level connectivity of LL. (**c**) Mean asymmetry index of G1 in macaques. (**d**) Normalized (Cohen's d) asymmetry of G1 in macaques and humans aligned to macaque's surface. Purple indicates leftward asymmetry, whereas yellow indicates rightward asymmetry. (**e**) Similarity of normalized asymmetry of G1 between humans and macaques. (**f**) The details of how the human Cole-Anticevic network atlas is projected to the macaque surface

*Figure 4 continued on next page*

*Figure 4 continued*

can be seen in the Methods. Bold colors indicate human mean cohen's D values in a given network and pastel colors indicate macaque mean cohen's D values in a given network. Networks are ranked from leftward (language) to rightward asymmetry (frontoparietal) along the intra-hemispheric principal gradient in humans for comparison.

The online version of this article includes the following figure supplement(s) for figure 4:

**Figure supplement 1.** Macaque versus human region-wise difference maps based on the normalized (Cohen's d) asymmetry of G1.

Humans and macaques showed an opposite direction of asymmetry in auditory, orbito-affective, and secondary visual networks. For the inter-hemispheric FC pattern, macaques and humans showed only subtle differences.

## Functional decoding along the normalized asymmetry of G1 (Figure 5)

Finally, we investigated the relationship between patterns of asymmetry of functional organization in humans and task-based meta-analytic functional activations. To do so, we projected meta-analytical fMRI activation maps (*Yarkoni et al., 2011*) along the normalized (Cohen's d) asymmetry of G1 (*Figure 5*). Our choice for the 24 cognitive domain terms were consistent with prior literature (*Margulies et al., 2016*). Here, we calculated the weighted score by activation z-score (parcels where activation z-score was greater than 0.5) multiplied by the normalized asymmetry, suggesting leftward to rightward preference, seen from top to bottom of the y-axis of *Figure 5*. Language, semantics, and reading domains were associated with leftward hemispheric preference, whereas cognitive control, inhibition, and working memory were associated with rightward hemispheric preference. For the asymmetry of the inter-hemispheric FC gradient, we observed a similar pattern of association (*Figure 5—figure supplement 1*). This indicates that patterns of asymmetry in functional organization also align with task-based activations consistently reported in the literature.

## Robustness analyses

Complementing our main AI calculation (L-R), we additionally used AI_norm (L-R)/(L+R), with rescaling the distribution of gradients to positive values, to explore whether our results were robust with respect to AI calculation (*Figure 2—figure supplement 3*). We found that for G1, asymmetric effects were highly correlated with the main asymmetric effects (Spearman $r_{intra\text{-}hemisphere}$ = 0.851, $P_{uncorrected}$ <0.001; $r_{inter\text{-}hemisphere}$ = 0.863, $P_{uncorrected}$ <0.001). Significant correlation was also found in G2 (Spearman $r_{intra\text{-}hemisphere}$ = 0.681, $P_{uncorrected}$ <0.001; $r_{inter\text{-}hemisphere}$ = 0.228, $P_{uncorrected}$ = 0.002) and in G3 (Spearman $r_{intra\text{-}hemisphere}$ = 0.795, $P_{uncorrected}$ <0.001; $r_{inter\text{-}hemisphere}$ = 0.879, $P_{uncorrected}$ <0.001).

To test the robustness of our findings with respect to the parcellation approach, we employed Desikan-Killiany atlas [1] to generate the asymmetry of functional gradients. This is a symmetric atlas containing 34 parcels per hemisphere. Overall, for the intra-hemispheric pattern G1 showed similar hemispheric patterns as observed in our main results when using the Desikan-Killiany atlas. In particular, the posterior cluster between middle and superior temporal gyrus and Broca's area showed leftward asymmetry, whereas dorsolateral prefrontal regions showed rightward asymmetry (*Figure 2—figure supplement 4*). However, we observed more details are shown in multi-modal parcellation. Similar patterns were observed in inter-hemispheric asymmetry of functional organization when using the Desikan-Killiany atlas.

We also used an additional sample (UK Biobank, UKB) to verify whether asymmetry of functional organization is present in other samples. We included 34,830 subjects' imaging data in UKB with good quality. After computing Cohen's d of asymmetric effects in UKB, to account for differences in sample size, we performed a group level correlation between HCP with UKB (*Figure 2—figure supplement 5*). We observed a high correlation between the LL functional gradient between HCP and UKB (Spearman $r_{G1}$ = -0.588, $P_{uncorrected}$ <0.001; Spearman $r_{G2}$ = 0.309, $P_{uncorrected}$ <0.001; Spearman $r_{G3}$ = 0.773, $P_{uncorrected}$ <0.001). Thus, we flipped UKB LL G1 direction to make it more consistent with HCP (now $r_{G1}$ = 0.588, $P_{uncorrected}$ <0.001). For the G1 of the intra-hemispheric FC pattern, we observed a correlation with our findings in the HCP sample (Pearson $r_{intra\text{-}hemisphere}$ = 0.592, $P_{spin}$ <0.001; Pearson $r_{inter\text{-}hemisphere}$ = 0.384, $P_{spin}$ <0.001). All the networks showed significant asymmetry in UKB. However, we found that language and default mode networks showed leftward asymmetry (as in HCP), but the frontoparietal network did not show rightward asymmetry.

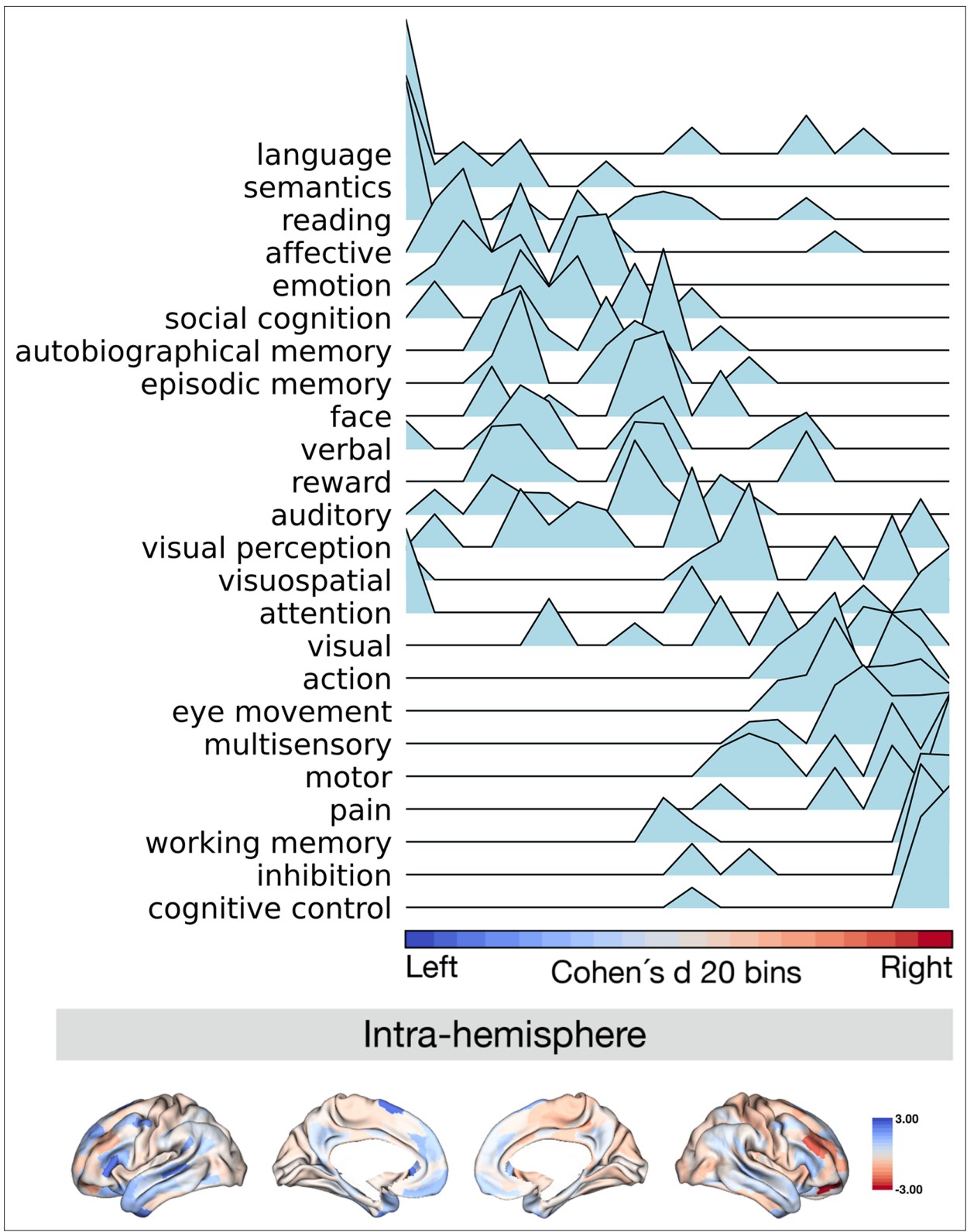

**Figure 5.** Projection of meta-analytical task-based function along normalized asymmetry of G1 (intra-hemisphere). The 20 bins were generated by normalized (Cohen's d) asymmetry of G1 in humans. Cool color indicates regions showing leftward dominance and warm color indicates regions showing rightward dominance. The order of the terms of the y-axis was generated by the weighted score of activation (z-score >0.5) * normalized asymmetry.

*Figure 5 continued on next page*

*Figure 5 continued*

The online version of this article includes the following figure supplement(s) for figure 5:

**Figure supplement 1.** Projection of meta-analytical task-based function along normalized (Cohen's d) asymmetry of G1 (inter-hemisphere).

Evaluating possible effects due to the parcellation scheme used, we studied differences of the mean rsfMRI connectome along the first gradient at the vertex level. We used 100 random subjects, as we had the data mapped to a symmetric template (fs_LR_32 k), which indicated that each vertex has a symmetric counterpart in the right hemisphere. Our results show left-right asymmetry being language/default mode-visual-frontoparietal vertices, which is consistent with the main results of the parcel-based approach (*Figure 2—figure supplement 6*).

To evaluate potential downstream effects of alignment to our results, we compared the gradient asymmetry with Procrustes alignment to the gradient without alignment. This resulted in virtually identical results for the HCP sample ($r_{\text{intra-hemisphere}} = 0.956$, $r_{\text{inter-hemisphere}} = 0.843$, *Figure 2—figure supplement 7*). At the same time, comparing unaligned and aligned gradients in the UKB sample, we found that the alignment improved the similarity to the pattern observed in HCP (aligned $r_{\text{intra-hemisphere}} = 0.592$, non-aligned $r_{\text{intra-hemisphere}} = 0.487$, aligned $r_{\text{inter-hemisphere}} = 0.384$, non-aligned $r_{\text{inter-hemisphere}} = 0.162$, *Figure 2—figure supplement 8*).

Moreover, to overcome potential normalization biases associated with creating one gradient for each hemisphere, we performed an alternative analysis to create a gradient of the left and right hemisphere together. This assumes that regions with similar connectivity profiles have comparable loading in the gradient framework. Indeed, along the principal gradient, the observed normalized asymmetric map was highly similar to the non-normalized map used in the main analyses for the intra-hemispheric (Pearson $r=0.956$) and inter-hemispheric (Pearson $r=0.531$) asymmetry patterns (*Figure 2—figure supplement 9*). It is possible the difference between intra- and inter-hemispheric correspondence relates to more global differences in strength of connectivity comparing LR to RL FC, as reported also in the article (*Raemaekers et al., 2018*) resulting in more widespread differences between inter-hemispheric patterns of both embedding procedures.

Finally, we also set the RR FC gradients as reference for our analyses, the first three of which explained 22.8, 18.8, and 15.9% of the total variance. We aligned each individual to this reference (*Figure 2—figure supplement 10*). It suggested all results were virtually identical (Pearson $r_{\text{intra G1}}=0.989$, $r_{\text{intra G2}}=0.939$, $r_{\text{intra G3}}=0.987$, $r_{\text{inter G1}}=0.979$, $r_{\text{inter G2}}=0.960$, $r_{\text{inter G3}}=0.990$, all $P_{\text{spin}} <0.001$).

## Discussion

In this study, we investigated the extent to which human cortical functional organization is asymmetric using a gradient-based approach. We assessed whether genetic factors shape such asymmetry and evaluated whether patterns of asymmetry are phylogenetically conserved between humans and non-human primates (macaques). We found that the principal gradient revealed hemispheric differences in most cortical regions, excluding the visual cortex. The language network and default-mode network showed the most leftward asymmetry while the frontoparietal network showed the most rightward asymmetry. The observed asymmetry of functional organization along the principal gradient was heritable. At the same time, regions with high asymmetry showed variable heritability. This may suggest that asymmetry in functional organization reflects both heritable and experience-dependent factors. Although the difference in left and right hemispheric functional organization was not significant along the principal functional gradient in a sample of macaques, the inter-hemispheric asymmetric pattern was comparable to the asymmetry pattern observed in humans indicating phylogenetic conservation. Notably, both the language and frontoparietal networks showed a higher leftward asymmetry in humans relative to macaques, indicating cross-species differences in asymmetry of specific transmodal functional networks. Decoding task-based functional activations along the asymmetry axis of the principal gradient, we observed that regions with a leftward preference were associated with language, autobiographical memory, and social cognition domains, whereas those with a rightward preference included cognitive control, working memory, and inhibition. In sum, our study shows the asymmetry of functional organization is, in part, heritable in humans and phylogenetically conserved in humans and macaques. At the same time, we observed that asymmetry of regions linked to higher-order cognitive functions such as language and cognitive control showed marked differences between

humans and macaques and variable heritability in humans, possibly reflecting an evolutionary adaptation allowing for experience-dependent specialization.

By studying asymmetry in functional organization using a gradient approach, we have extended previous studies reporting asymmetric functional connectivity. Indeed, although the functional organization of the cerebral cortex has a largely symmetric pattern, it also shows subtle differences between hemispheres (*Liang et al., 2021*; *Gonzalez Alam et al., 2022*; *Iturria-Medina et al., 2011*; *Sun et al., 2017*). For the intra-hemispheric asymmetry gradients, we found that regions belonging to the language network showed the strongest leftward preference along the principal gradient axis. This indicates that their functional connectivity profiles were more similar to the default mode, relative to their right-hemispheric counterparts. Conversely, ventral multimodal networks were closer to the transmodal apex of the principal gradient in the right hemisphere, relative to their homologues in the left hemisphere. As such, our observations suggest that key transmodal regions, part of the language and control networks, show organizational preference to either the left or right hemisphere. Anterior lateral default mode subnetworks have been shown to uniquely exhibit positive connectivity to the language network (*Gordon et al., 2020*), possibly leading to increased gradient loadings of the language network in the left hemisphere, placing them closer to the default regions along the principal gradient in the left hemisphere relative to the right. Conversely, the transmodal frontoparietal network was located at the apex of rightward preference, possibly suggesting a rightward lateralization of cortical regions associated with attention and control and 'default' internal cognition (*Corbetta and Shulman, 2002*; *Smallwood et al., 2021*). The observed dissociation between language and control networks is also in line with previous work suggesting an inverse pattern of language and attention between hemispheres (*Karolis et al., 2019*; *Zago et al., 2016*). Such patterns may be linked to inhibition of corpus callosum (*Cook, 1984*), promoting hemispheric specialization. It has been suggested that such inter-hemispheric connections set the stage for intra-hemispheric patterns related to association fibers (*Gazzaniga, 2000*). Future research may relate functional asymmetry directly to asymmetry in underlying structure to uncover how different white-matter tracts contribute to asymmetry of functional organization.

We furthermore investigated whether such individual variations in asymmetry of functional organization could be attributed to genetic factors. To do so, we performed heritability analysis enabled by the twin design of the Human Connectome Project (*Van Essen et al., 2013*). Previous work indicated that brain structure including cortical thickness, surface area, and white matter connection (*Kong et al., 2018*; *Sha et al., 2021*; *Zhong et al., 2021*), as well as functional connectome organization (*Colclough et al., 2017*; *Ge et al., 2017*) are heritable. Our twin-based heritability analyses revealed heritable asymmetry of the principal functional gradient in the entire cortex, excluding visual cortex. At the same time, studying the association between heritability and asymmetry patterns we observed mixed results. Although we observed that the language-related area PSL showed the highest heritability, the highly asymmetric area 44 (Broca's area) showed the lowest heritability. This may reflect a differential (dorsal and ventral) pathway of language development in the frontal and temporal lobe, where the dorsal pathway to the inferior frontal gyrus matures at later stages in development (*Brauer et al., 2013*). For example, previous work found that temporal language areas showed high heritability of cortical thickness asymmetry (*Sha et al., 2021*) and white matter connection asymmetry (*Zhong et al., 2021*) but frontal language areas did not. Such posterior-anterior differences may be due to developmental factors or axes of stability versus plasticity in the cortex (*García-Cabezas et al., 2017*). A case study of an individual born without a left temporal lobe found that frontal language areas in the left hemisphere did not emerge in the absence of temporal language areas in the left hemisphere, and that language functions instead relied on the right hemispheric functional network (*Tuckute et al., 2022*). It is thus possible that Broca's area may mature after more posterior language regions in hierarchical fashion, which may be related to decreasing heritability in frontal language areas (i.e. more influenced by developmental and/or environmental factors). Recent work suggests that asymmetric patterning of brain structure and function are largely determined prenatally and unaffected by preterm birth (*Williams et al., 2021*). In neonates, asymmetric patterns were largely observed in primary and unimodal areas, whereas association regions were largely symmetric. Thus, asymmetry in association regions may be more experience-dependent. One focus of future work could thus be to evaluate the development of asymmetry in functional organization. Moreover, by means of GWAS approaches, it may also

be possible to get more insight in specific genes and associated processes involved in functional asymmetry.

Evaluating the correspondence of asymmetry of functional organization between humans and macaques, by aligning the human gradients to the macaque gradient space (*Xu et al., 2020*), we observed a similarity between asymmetry of the principal gradient in both species in case of intra-hemispheric connectivity. This indicates functional asymmetry of within-hemispheric connectivity may be conserved across primates. At the same time, we found that language, default mode, and frontoparietal networks showed qualitatively more asymmetry in humans (human >macaque). These findings may support the notion that though asymmetry is a phenomenon existent across different primates, regions involved in higher order cognitive functions in humans are particularly asymmetric. Previous work studying asymmetry in white matter structure in primates found that humans showed more leftward arcuate fasciculus volume and surface relative to macaques (*Eichert et al., 2019*). The arcuate fasciculus is a white matter tract implicated in language functions by connecting Broca's and Wernicke's areas (*Geschwind, 1970*). Moreover, by comparing humans, macaques, and chimpanzees (*Eichert et al., 2020*), evolutionary modifications to this tract in humans relative to other primates have been reported, possibly derived from auditory pathways (*Balezeau et al., 2020*). At the same time, other structural studies have also observed leftward asymmetry of language areas in chimpanzees, indicating that asymmetry of language-regions per se may not be a human-specific feature (*Spocter et al., 2020*; *Xiang et al., 2020*). Fittingly, there are no significant differences of thickness and area asymmetry between humans and chimpanzees in superior temporal lobe (*Xiang et al., 2020*). Studying the endocranial shape of humans and non-human primates, temporal and occipital cortices showed local differences in asymmetry across species, and much more variability in humans relative to non-human primates (*Neubauer et al., 2020*). This suggests that whereas brain asymmetry is a phenomenon observed throughout mammals (*Assaf et al., 2020*), specific nuances may relate to species-specific behavioral and cognitive differences. Future research could assess asymmetry of brain organization in other primates, and relate observed differences in functional organization to those in white matter structure.

Although overall intra- and inter-hemispheric connectivity showed a strong spatial overlap in humans, we also observed various differences between the metrics across our analysis. For example, although we found both intra- and inter-hemispheric differences in gradient organization to be heritable, only for the former was a correspondence between the degree of asymmetry and heritability found. Similarly, comparing human and macaques, we only observed conservation of spatial patterning of asymmetry was conserved for intra-hemispheric connections. Whereas intra-hemispheric asymmetry relates to association fibers, commissural fibers underlie inter-hemispheric connections (*Tzourio-Mazoyer, 2016*). It has been suggested that there is a trade-off within and across mammals of inter- and intra-hemispheric connectivity patterns to conserve the balance between gray and white-matter (*Assaf et al., 2020*). Consequently, differences in asymmetry of both ipsi- and contralateral functional connections may be reflective of adjustments in this balance within and across species. Secondly, previous research studying intra- and inter-hemispheric connectivity and associated asymmetry, has indicated a developmental trajectory from inter- to intra-hemispheric organization of functional brain connectivity, varying from unimodal to transmodal areas (*Friederici et al., 2011*; *Szaflarski et al., 2006*). It is thus possible that a reduced correspondence of asymmetry and heritability in humans, as well as lack of spatial similarities between humans and macaques for inter-hemispheric connectivity may be due to the age of both samples (young adults in humans, adolescents in macaques). Further research could study inter- and intra-hemispheric asymmetry in functional organization as a function of development in both species, to further disentangle heritability and cross-species conservation and adaptation.

The functional relevance of asymmetry along the sensory-transmodal axis was evaluated in the human brain by projecting meta-analytical task-based coactivations along asymmetric effects of the functional principal gradient. In line with our expectations based on the distribution of asymmetry within functional networks, we found that task-based activations associated with language processing leaned leftward while task-based activations associated with executive functions leaned rightward, specifically in the intra-hemispheric pattern. This suggests that lateralized functions supported by the brain's asymmetry have functional relevance (especially higher order cognitive functions such as language and executive control). Indeed, related work has shown a direct link between asymmetry and semantic and visual recognition skills (*Gonzalez Alam et al., 2022*), suggesting that asymmetry

of individuals relates to variation in behavioral performance in these domains (*Gonzalez Alam et al., 2022*; *Gonzalez Alam et al., 2019*). Our observation of asymmetry of language versus executive functions may also be in line with notions of differential axes of asymmetry, dissociating symbolic/language, emotion, perception/action, and decision functional axes (*Karolis et al., 2019*). The asymmetry of principal functional gradient in humans and macaques showed a divergence along these axes, possibly indicating cross-species variability within the lateralization archetypes in primates. Notably, left hemispheric language lateralization is enabled throughout language development while right hemispheric language activation declines systematically with age (*Olulade et al., 2020*). Therefore, future research may focus on studying how the lateralization of human behavior is shaped by development and aging and how this may impact function and behavior.

Although we showed asymmetry in functional organization, there are various technical and methodological aspects to be considered. In the current work, we used the MMP (*Glasser et al., 2016*) for surface-based human fMRI data. A previous study used the atlas of intrinsic connectivity of homotopic areas (*Xiang et al., 2020*) (AICHA, https://www.gin.cnrs.fr/en/tools/aicha) for voxel-based fMRI data (*Liang et al., 2021*). In line with the results of that study, we found similar intra-hemispheric differences in functional gradients. Extending that work we additionally used the DK atlas (*Desikan et al., 2006*), which is often used in structural asymmetry studies (*Sha et al., 2021*; *Postema et al., 2019*). We again found asymmetric patterns, with a rightward dorsal frontal lobe and leftward posterior superior temporal lobe. The other temporal regions, having leftward or rightward asymmetry using MMP, showed no or less asymmetry using the DK atlas. Possibly, such subtle differences are not captured by the DK atlas, with only 34 cortical parcels per hemisphere. Evaluating the consistency of functional asymmetry across different datasets, we found that HCP (n=1014) and UKB (n=34,604) showed consistent leftward asymmetric functional organization in the language and default mode networks but no consistent rightward asymmetry of the frontoparietal network. Such differences may be due to technical differences between the datasets (*Xifra-Porxas et al., 2021*). However, it may also reflect sample specific differences in asymmetry. Indeed, whereas the HCP sample consists of young-adults with an age-range of 22–37 years, the UKB has a comparatively older and wider age range (from 40 to more than 70 years). Thus, it is possible the observed differences in the frontoparietal network are directly related to age-related asymmetry effects (*Olulade et al., 2020*). Due to the small sample size of macaques, it is important to be careful when interpreting our observations regarding the associated asymmetry, and its relation to patterns observed in humans. Therefore, further study is needed to evaluate the asymmetry patterns in macaques using large datasets (*Milham et al., 2018*; *Messinger et al., 2021*).

To conclude, we investigated the genetic and phylogenetic basis of asymmetry of large-scale functional organization. We observed that the principal (unimodal-transmodal) gradient (*Margulies et al., 2016*) is asymmetric, with regions involved in language showing leftward organization and regions associated with executive function showing rightward organization. This asymmetry was heritable and, in the case of organization of intra-hemispheric connectivity, showed spatial correspondence between humans and macaques. At the same time, functional asymmetry was more pronounced in language networks in humans relative to macaques, suggesting adaptation. The current framework may be expanded by future research investigating the development and phylogeny of functional asymmetry as well as its neuroanatomical basis in healthy and clinical samples. This may provide important insights in individual-level brain asymmetry and its relation to human cognition.

## Materials and methods

The current research complies with all relevant ethical regulations as set by The Independent Research Ethics Committee at the Medical Faculty of the Heinrich-Heine-University of Duesseldorf (study number 2018–317).

### Participants

#### Humans

For the analyses in humans, we used the Human Connectome Project (HCP) S1200 data release (*Van Essen et al., 2013*). That release contains four sessions of resting state (rs) fMRI scans for 1206 healthy young adults and their pedigree information (298 monozygotic and 188 dizygotic twins as well as

720 singletons). We included individuals with a complete set of four fMRI scans that passed the HCP quality assessment (*Van Essen et al., 2013*; *Glasser et al., 2013*). Finally, our sample consisted of 1014 subjects (470 males) with a mean age of 28.7 years (range: 22–37).

For the replication, we employed the UKB dataset (application ID: 41655) including 34,830 subjects' imaging data. Details on data processing and acquisition can be found in the UKB Brain imaging documentation (https://biobank.ctsu.ox.ac.uk/crystal/crystal/docs/brain_mri.pdf). Briefly, resting-state imaging data was motion corrected, intensity normalized, high-pass temporally filtered, and further denoised using the ICA-FIX pipeline, all implemented in FSL. MPM parcellation was warped to subject-space based on the high-resolution T1-weighted anatomical image. Individual warping parameters were applied to map the MPM parcellation to the functional space following T1-rsfMRI alignment. The age range of the UKB sample was from 40 to more than 70 years.

### Macaques
We selected rhesus macaque monkeys' rs-fMRI data from the non-human primate (NHP) consortium PRIME-DE (http://fcon_1000.projects.nitrc.org/indi/indiPRIME.html) from Oxford. The full dataset consisted of 20 rhesus macaque monkeys (macaca mulatta) scanned on a 3T with a 4-channel coil (*Noonan et al., 2014*). The rs-fMRI data were collected with 2 mm isotropic resolution, TR = 2 s, 53.3 mins (1600 volumes). Details can be seen in *Xu et al., 2020*. Nineteen macaques with successful preprocessing and surface reconstruction were included in the current study (all males, age = 4.01 ± 0.98 years, weight = 6.61 ± 2.04 kilograms).

Macaque data were preprocessed with an HCP-like pipeline (*Xu et al., 2015*), described elsewhere (*Xu et al., 2020*). In brief, it included temporal compression, motion correction, 4D global scaling, nuisance regression using white matter (WM), cerebrospinal fluid (CSF), and Friston-24 parameter models, bandpass filtering (0.01–0.1 Hz), detrending, and co-registration to the native anatomical space. The data were then projected to the native midcortical surface and smoothed along the surface with FWHM = 3 mm. Finally, the preprocessed data were down-sampled to the surface space (with resolution of 10,242 vertices in each hemisphere).

## Parcellations
### Multimodal parcellation and Cole-Anticevic network
We used multimodal parcellation (MMP) of 360 areas (180 per hemisphere) for humans (*Glasser et al., 2016*). This atlas has been generated using the gradient-based parcellation approach with similar gradient ridges presenting in roughly corresponding locations in both hemispheres, which is suitable for studying asymmetry across homologous parcels. Additionally, based on MMP, we used the Cole-Anticevic Brain-wide Network Partition (CA network), which includes in total 12 functional networks (*Ji et al., 2019*).

### Desikan-Killiany atlas
To ensure our results were reliable we repeated the analysis in humans using a different brain atlas. The Desikan-Killiany atlas (*Desikan et al., 2006*) contains 34 cortical parcels per hemisphere in humans and has high correspondence across two hemispheres.

### Markov parcellation
For the macaques, we used 91 cortical areas per hemisphere in the Markov M132 architectonic parcellation (*Markov et al., 2014*). This directed and weighted atlas is generated based on the connectivity profiles. The 91-area parcellation in macaques is valuable for comparison with connectivity analyses in humans.

## Functional connectivity
All rs-fMRI data underwent HCP's minimal preprocessing (*Glasser et al., 2013*) and were coregistered using a multimodal surface matching algorithm (MSMAll) (*Robinson et al., 2014*) to the HCP template 32 k_LR surface space. The template consists of 32,492 total vertices per hemisphere (59,412 excluding the medial wall). Cortical time series were averaged within a previously established multi-modal parcellation schemes: for humans the 360-parcel Glasser atlas (180 per hemisphere) (*Glasser et al., 2016*) and the 182-parcel Markov atlas (91 per hemisphere) for macaques (*Markov*

*et al., 2014*). To compute the functional connectivity (FC), time-series of cortical parcels were correlated pairwise using the Pearson product moment and then Fisher's z-transformed in human and macaque data, separately. Individual FC maps were also averaged across four different rs-fMRI sessions for humans ([LR1], [LR2], [RL1], and [RL2]). We computed the FC in four different patterns, both for human and macaque data: FC within the left and right hemispheres (LL intra-hemisphere, RR intra-hemisphere), from the left to right hemisphere (LR inter-hemisphere) and from the right to left hemisphere (RL, inter-hemisphere).

## Connectivity gradients

Next we employed the nonlinear dimensionality reduction technique (*Margulies et al., 2016*) to generate the group level gradients of the mean LL FC across individuals. We then set the group-level gradients as the template and aligned each individual gradient with Procrustes rotation to the template. Finally, the comparative individual functional gradients of each FC pattern were assessed. All steps were accomplished in the Python package Brainspace (*Vos de Wael et al., 2020*). In brief, the algorithm estimates a low-dimensional embedding from a high-dimensional affinity matrix. Along these low-dimensional axes, or gradients, cortical nodes that are strongly interconnected, by either many suprathreshold edges or few very strong edges, are closer together. Nodes with little connectivity similarly are farther apart. Regions having similar connectivity profiles are embedded together along the gradient axis. The name of this approach, which belongs to the family of graph Laplacians, is derived from the equivalence of the Euclidean distance between points in the diffusion embedded mapping (*Coifman et al., 2005*; *Margulies et al., 2016*; *Vos de Wael et al., 2020*). It is controlled by a single parameter α, which controls the influence of the density of sampling points on the manifold ($\alpha=0$, maximal influence; $\alpha=1$, no influence). On the basis of the previous work (*Margulies et al., 2016*), we followed recommendations and set $\alpha=0.5$, a choice that retains the global relations between data points in the embedded space and has been suggested to be relatively robust to noise in the covariance matrix.

The input of the analysis was the FC matrix, which was cut off at 90% similar to previous studies (*Margulies et al., 2016*). The current study selected the first three FC LL gradients (G1, G2, and G3) that explained 24.1, 18.4, and 15.1% of total variance in humans, as well as 18.9, 15.2, and 12.8% of total variance in macaques.

## Asymmetry index

To quantify the left and right hemisphere differences, we chose left-right as the asymmetry index (AI) (*Liang et al., 2021*; *Raemaekers et al., 2018*). In addition, we also calculated the normalized AI with the following formula, (left-right)/(left +right), which is usually used in structural studies to verify whether there is a difference between unnormalized AI and normalized AI. For the intra-hemispheric pattern, the AI was calculated using LL-RR. A positive AI-score meant that the hemispheric feature dominated leftwards, while a negative AI-score dominated rightwards. For the inter-hemispheric pattern we used LR-RL to calculate the AI. Notably, we added 'minus' to the AI scores or Cohen's d scores in the figures in order to conveniently view the lateralization direction.

## Heritability analysis

To map the heritability of functional gradient asymmetry in humans, we used the Sequential Oligogenic Linkage Analysis Routines (SOLAR, v8.5.1b) (*Almasy and Blangero, 1998*). In brief, heritability indicates the impact of genetic relatedness on a phenotype of interest. SOLAR uses maximum likelihood variance decomposition methods to determine the relative importance of familial and environmental influences on a phenotype by modeling the covariance among family members as a function of genetic proximity (*Valk et al., 2020*; *Almasy and Blangero, 1998*). Heritability (i.e. narrow-sense heritability $h^2$) represents the proportion of the phenotypic variance ($\sigma^2_p$) accounted for by the total additive genetic variance ($\sigma^2_g$), that is $h^2 = \sigma^2_g / \sigma^2_p$. Phenotypes exhibiting stronger covariances between genetically more similar individuals than between genetically less similar individuals have higher heritability. In this study, we quantified the heritability of asymmetry of functional gradients. We added covariates to our models including age, sex, age$^2$, and age ×sex.

## Alignment of humans to macaques

To phylogenetically map the asymmetry of functional gradients across macaques and humans, we transformed the human gradients to macaque cortex surface based on a functional joint alignment technique (*Xu et al., 2020*). This method leverages advances in representing functional organization in high-dimensional common space and provides a transformation between human and macaque cortices, also previously used in *Valk et al., 2020*; *Valk et al., 2022*.

In the present study, we aligned Cohen's d of the human asymmetry index to the macaque surface. Cohens' d explains the effect size of the asymmetry index. Following the joint alignment, we further computed the Spearman correlation between macaques and humans to evaluate the similarity in asymmetric patterns of the functional gradients. Finally, we compared Cohen's d between macaques and humans and summarized the results with Markov parcellation (*Markov et al., 2014*). To illustrate our findings at the functional network level, we projected human networks (*Ji et al., 2019*) on the macaque surface.

## NeuroSynth meta-analysis

To evaluate the association of function decoding and asymmetry of the principal gradient, we projected the meta-analytical task-based activation along the normalized asymmetry (Cohen's d) of G1. Our choice for the 24 cognitive domain terms were consistent with (*Margulies et al., 2016*). The activation database we used for meta-analyses was the Neurosynth V3 database (*Yarkoni et al., 2011*). The surface-based V3 database is available in the github depository (data availability). In the present study, to look at how the right hemisphere and left hemisphere decode functions separately, the leftward normalized asymmetry was put on and the rightward normalized asymmetry was put on the right hemisphere. Other regions became zero. We generated 20 bins along the normalized asymmetry averagely (5% per bin). Thus, each function term had a mean activation z-score per bin. To assess how much the function term was leftward or rightward lateralized, we calculated a weighted score by mean activation (where activation z-score greater than 0.5) multiplied by normalized asymmetry. We roughly regarded this score as the lateralization level. The order of the function terms generated by this calculation reflected the left-right lateralization dominance axis.

## Data availability

All human data analyzed in this manuscript were obtained from the open-access HCP young adult sample (https://www.humanconnectome.org/), UK Biobank (https://www.ukbiobank.ac.uk/). Macaque data came from PRIME-DE (http://fcon_1000.projects.nitrc.org/indi/indiPRIME.html). Gradient analyses and visualization were performed using the Python package Brainspace (*Vos de Wael et al., 2020*) (https://brainspace.readthedocs.io/en/latest/index.html). Heritability analyses were performed using Solar Eclipse 8.5.1b (https://www.solar-eclipse-genetics.org). Task-based function association analyses were based on NeuroSynth (*Yarkoni et al., 2011*) (https://neurosynth.org/). Full statistical scripts can be found at https://github.com/CNG-LAB/cngopen/tree/main/asymmetry_functional_gradients (copy archived at swh:1:rev:07d4a1a03267dac12ac8bfbccc8e09049cac9f31;path=/asymmetry_functional_gradients; *Bayrak et al., 2022*).

## Acknowledgements

We would like to thank the various contributors to the open access databases that our data was downloaded from. Funding: HCP data were provided by the Human Connectome Project, Washington University, the University of Minnesota, and Oxford University Consortium (Principal Investigators: David Van Essen and Kamil Ugurbil;1U54MH091657) funded by the 16 NIH Institutes and Centers that support the NIH Blueprint for Neuroscience Research; and by the McDonnell Center for Systems Neuroscience at Washington University. Additional personnel support provided by the Center for the Developing Brain at the Child Mind Institute, as well as NIMH R01MH081218, R01MH083246, and R21MH084126. Project support is also provided by the NKI Center for Advanced Brain Imaging (CABI), the Brain Research Foundation (Chicago, IL), and the Stavros Niarchos Foundation. This study was supported by the Deutsche Forschungsgemeinschaft (DFG, EI 816/21–1), the National Institute of Mental Health (R01-MH074457), the Helmholtz Portfolio Theme "Supercomputing and Modeling for the Human Brain'' and the European Union's Horizon 2020 Research and Innovation Program under

Grant Agreement No. 785,907 (HBP SGA2). SLV was supported by Max Planck Gesellschaft (Otto Hahn award). BCB acknowledges support from the SickKids Foundation (NI17-039), the National Sciences and Engineering Research Council of Canada (NSERC; Discovery-1304413), CIHR (FDN154298), Azrieli Center for Autism Research (ACAR), an MNI-Cambridge collaboration grant, and the Canada Research Chairs program. Last, this work was funded in part by Helmholtz Association's Initiative and Networking Fund under the Helmholtz International Lab grant agreement InterLabs-0015, and the Canada First Research Excellence Fund (CFREF Competition 2, 2015–2016) awarded to the Healthy Brains, Healthy Lives initiative at McGill University, through the Helmholtz International BigBrain Analytics and Learning Laboratory (HIBALL), including SLV and BCB. BW was supported by the International Max Planck Research School on Neuroscience of Communication: Function, Structure, and Plasticity (IMPRS NeuroCom).

## Additional information

### Funding

| Funder | Grant reference number | Author |
| --- | --- | --- |
| Max-Planck-Gesellschaft | | Sofie L Valk |
| Sick Kids Foundation | NI17-039 | Boris C Bernhardt |
| Natural Sciences and Engineering Research Council of Canada | Discovery-1304413 | Boris C Bernhardt |
| Canadian Institutes of Health Research | FDN154298 | Boris C Bernhardt |
| Azrieli Center for Autism Research | | Boris C Bernhardt |
| Canada First Research Excellence Fund | | Boris C Bernhardt Sofie L Valk |
| International Max Planck Research School on Neuroscience of Communication: Function, Structure, and Plasticity | | Bin Wan |

The funders had no role in study design, data collection and interpretation, or the decision to submit the work for publication.

### Author contributions

Bin Wan, Conceptualization, Data curation, Formal analysis, Supervision, Validation, Visualization, Methodology, Writing - original draft, Project administration, Writing – review and editing; Şeyma Bayrak, Validation, Visualization, Methodology, Writing – review and editing; Ting Xu, Richard AI Bethlehem, Data curation, Writing – review and editing; H Lina Schaare, Methodology, Writing – review and editing; Boris C Bernhardt, Funding acquisition, Methodology, Writing – review and editing; Sofie L Valk, Conceptualization, Data curation, Supervision, Funding acquisition, Validation, Methodology, Project administration, Writing – review and editing

### Author ORCIDs

Bin Wan http://orcid.org/0000-0001-9077-3354
H Lina Schaare http://orcid.org/0000-0003-4259-0793
Richard AI Bethlehem http://orcid.org/0000-0002-0714-0685
Boris C Bernhardt http://orcid.org/0000-0001-9256-6041
Sofie L Valk http://orcid.org/0000-0003-2998-6849

### Ethics

The current research complies with all relevant ethical regulations as set by The Independent Research Ethics Committee at the Medical Faculty of the Heinrich-Heine-University of Duesseldorf (study number 2018-317).

### Decision letter and Author response

Decision letter https://doi.org/10.7554/eLife.77215.sa1
Author response https://doi.org/10.7554/eLife.77215.sa2

---

## Additional files

### Supplementary files

• Supplementary file 1. Summary of asymmetry index and heritability of G1.
• MDAR checklist

### Data availability

All human data analyzed in this manuscript were obtained from the open-access HCP young adult sample (https://www.humanconnectome.org/), UK Biobank (https://www.ukbiobank.ac.uk/). Macaque data came from PRIME-DE (http://fcon_1000.projects.nitrc.org/indi/indiPRIME.html). Gradient analyses and visualization were performed using the Python package Brainspace (Vos de Wael et al., 2020) (https://brainspace.readthedocs.io/en/latest/index.html). Heritability analyses were performed using Solar Eclipse 8.5.1b (https://www.solar-eclipse-genetics.org). Task-based function association analyses were based on NeuroSynth (Yarkoni et al., 2011) (https://neurosynth.org/). Full statistical scripts can be found at https://github.com/CNG-LAB/cngopen/tree/main/asymmetry_functional_gradients (copy archived at swh:1:rev:07d4a1a03267dac12ac8bfbccc8e09049cac9f31;path=/asymmetry_functional_gradients).

The following previously published dataset was used:

| Author(s) | Year | Dataset title | Dataset URL | Database and Identifier |
|-----------|------|---------------|-------------|-------------------------|
| HCP S1200 | 2018 | 1200 Subjects Data Release | https://db.humanconnectome.org/ | ConnectomeDB, S1200 |

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
