## [Editor Report]

This is a valuable paper investigating hemispheric asymmetries in brain functional connectivity. The authors quantify this asymmetry using a solid methodology that capitalises on recent developments in functional gradients, and they further ask if these asymmetries are heritable and how they compare between humans and macaque monkeys. The results suggest a genetic underpinning of brain functional asymmetry, particularly in areas supporting unique human functions. These findings may help further our understanding of brain asymmetries.

---

## [Decision Letter]

**Decision letter after peer review:**

Thank you for submitting your article "Asymmetry of cortical functional hierarchy in humans and macaques suggests phylogenetic conservation and adaptation" for consideration by *eLife*. Your article has been reviewed by 2 peer reviewers, including Saad Jbabdi as the Reviewing Editor and Reviewer #1, and the evaluation has been overseen by Timothy Behrens as the Senior Editor.

Essential revisions:

There are two important points to address in your revisions:

1) The theoretical point raised by reviewer 1 regarding interpretation of localised changes in the embedding space. This seems to be a fundamental limitation of the method.

2) The point raised by reviewer 2 about insight gained by this study.*Reviewer #1 (Recommendations for the authors):*

The paper is very well written, the question is interesting, and the analyses are innovative. However, I do have concerns about the overall approach. My main concern is about looking at asymmetries in the low dimensional representation of connectivity. A secondary concern has to do with looking at the parcellated connectome. I explain these concerns in succession below.

The first concern is to me quite a fundamental issue: looking at connectivity in a low dimensional space, that of the laplacian eigenvectors. There are two issues with this. The first one, which is less important than the second, is that the authors have a reference embedding to which they align other embeddings using a procrustes method with no scaling. While the 3D embedding is still optimally representing the connectivity (because distances don't change under rotations), we can no longer look at one axis at a time, which is what the authors do when they look at G1. In this case, G1 is representative of the connectivity of the reference matrix (LL), but not the others.

But even if the authors only projected their matrices onto a single G1 dimension with no procrustes (and only sign flipping if necessary), there is still a major issue. One implicit assumption of this whole approach is that if there is a change in connectivity somewhere in the original matrix, the same "nodes" of the matrix will change in the embedding. This is not the case. Any change in the original matrix, even if it is a single edge, will affect the positions of all the nodes in the embedding. That is because the embedding optimises a global loss function, not a local one.

To make this point clear, consider the following toy example. Say we have 4 brain regions A,B,C,D. Let us say that we have the following connectivity:

In the Left Hemisphere: A-B-C-D

In the Right Hemisphere: A-B=C-D

So the connection between B and C is twice as strong in the right hemi, and everything else remains the same.

The low dimensional embedding of both will look like this:

Left:

… A … B ……. C … D …

Right

A… … … B … C … … … D

Note how B,C are closer to each other in the RIGHT, but also that A,D have moved away from each other because the eigenvector has to have norm 1.

So if we were to calculate an asymmetry index, we would say that:

A is higher on the LEFT

B is higher on the RIGHT

C is higher on the LEFT

D is higher on the RIGHT

So we have found asymmetry in all of our regions. But in fact the only thing that has changed is the connection between B and C.

This illustrates the danger of using a global optimisation procedure (like low-dim embedding) to analyse and interpret local changes. One has to be very careful.

My second concern is about interpreting the brain asymmetry as differences in connectivity, as opposed to differences in other things like regional size. The authors use a parcellated approach, where presumably the parcels are left-right symmetric. If one area is actually larger in one hemisphere than in the other, the will manifest itself in the connectivity values. To mitigate this, it may be necessary to align the two hemispheres to each other (maybe using spherical registration) using connectivity prior to applying the parcellation.

Figure 1. Please explain what "explained variance" means. The gradients represent a low dimensional version of the connectivity matrix. they are not explaining variance?*Reviewer #2 (Recommendations for the authors):*

Using recently-developed functional gradient techniques, this study explored human brain hemispheric asymmetry. The functional gradient is a hot technique in recent years and has been applied to study brain asymmetries in two papers of 2021. Compared to previous studies, the current study further evaluated the degree of genetic control (heritability) and evolutionary conservation for such gradient asymmetries by using human twin data and monkey's fMRI data. These investigations are of value and do provide interesting data. However, it suffers from a lack of specific hypotheses/questions/motivations underlying all kinds of analyses, and the rich observational or correlational results seem not to offer significant improvement of theoretical understanding about brain asymmetries or functional gradient. In addition, given the limited number of twins in HCP project (for a heritability estimation), the limited number of monkeys (20 monkeys), and the relatively poor quality of monkeys' resting functional MRI data, the results and conclusion should be taken cautiously. Below are the concerns and suggestions.

The gradient from resting-state functional connectome has been frequently used but mainly at the group level. The current study essentially applied the gradient comparison (i.e., gradient score) at the individual level. Biological interpretation for individual gradient score at the parcel level as well as its comparability between individuals and between hemispheres should be resolved. This is the fundamental rationale underlying the whole analyses.

Only the first three gradients are used but why? What about the fourth gradient? Specific theoretical interpretation is needed. At the individual level, is it ensured that the first gradients of all individuals correspond to each other? In this study, it is unclear whether we should or should not care about the G2 and G3. The results of G2 and G3 showed up randomly to some degree.

The intra-hemispheric gradient is institutive. However, it is hard to understand what the inter-hemispheric gradient means. From the data perspective, yes you can do such gradient comparison between the LR and RL connectome but what does this mean? Why should we care about such asymmetry? From the introduction to the discussion, the authors simply showed the data of inter-hemispheric gradients without useful explanation. This issue should be solved.

When aligning intra-hemispheric gradient, choosing averaged LL mode as the reference may introduce systematic bias towards left hemisphere. Such an issue also applies to LR-RL gradient alignment as well as cross-species gradient alignment. This methodological issue should be solved.

The sample size of monkey (i.e., 20) is far less than human subjects (> 1000). Such limitation raises severe concern on the validity of the currently observed gradient asymmetry pattern in the monkey group, as well as the similarity results with human gradient asymmetry pattern. Despite the marginal significance of G1 inter-hemisphere gradient between humans and monkeys, I feel overall there is no convincingly meaningful similarity between these two species. However, the authors' discussion and conclusion are largely based on strong inter-species similarity in such asymmetry. The conclusion of evolutionary conservation for gradient asymmetry, therefore, is not well supported by the results.

For human gradient asymmetry, only t values were provided; For monkey gradient asymmetry, only Cohen-d values were provided. These two should be provided for both species.

Figure 3b, it is hard to believe that such a scatter plot can reach a significant correlation of R>0.3. In addition, such a scatter plot does not match the text (i.e., correlation between the "absolute" AI and heritability)

Figure S3, why should we care about these cross-gradient correlations?

More detailed description for fMRI post-processing for functional connectome and gradient analyses could be added in the supplementary information.

DK atlas is not a good validation parcellation for a functional MRI study like this.

[Editors’ note: further revisions were suggested prior to acceptance, as described below.]

Thank you for resubmitting your work entitled "Heritability and cross-species comparisons of asymmetry of human cortical functional organization" for further consideration by *eLife*. Your revised article has been evaluated by Timothy Behrens (Senior Editor) and a Reviewing Editor.

The manuscript has been improved but there are some remaining issues that need to be addressed, as outlined below:

*Reviewer #1 (Recommendations for the authors):*

Thanks for your replies to my comments, and sorry for the delay in getting this response to you.

Regarding my first comment, i.e. interpretation of a change in position along the gradients: I am not sure I understand your reply. You agreed that it is difficult to interpret these changes, given that they can represent changes occurring outside the region where the change is reported, but then the analysis you have done does not address this concern. Instead, you calculated some other measure (which I am not sure what it is as it is not well described) and reported the asymmetry index using this new measure. If this new measure is more interpretable, then why do you need to use gradients? What information from the gradients is useful for the study of asymmetries? And how can we interpret changes in positions along the gradients? Simply saying that "interpretation for asymmetry of areas is under a global context" seems to me like sweeping the issue under the rug.

Regarding the issue of using the Procrustes to the template and how that makes the gradients a worse representation of connectivity for the non-template matrices: I don't understand the reply here either. What is meant by joint alignment and how exactly does this address my concern?

If I may add a couple of additional points:

– I said in my original review that this was a well-written paper, but it looks like the writing has gotten worse in this revised paper. I am not sure why that is, but I really invite the authors to re-read the paper, particularly the new sections/paragraphs, and ensure that the arguments make sense (and I don't just mean the English).

– Some of the captions are way too short. They are often comprised of just a few words, which is ok for a caption "title", but not for a caption. A caption needs to explain what is shown avoiding reference to the main text.

– The code provided is poorly organised and not documented. I encourage the authors to improve on that.

---

## [Author Response]

Reviewer #1 (Recommendations for the authors):The paper is very well written, the question is interesting, and the analyses are innovative. However, I do have concerns about the overall approach. My main concern is about looking at asymmetries in the low dimensional representation of connectivity. A secondary concern has to do with looking at the parcellated connectome. I explain these concerns in succession below.

We thank the Reviewer for the appreciation of our work and the insightful comments, which we have addressed below.

The first concern is to me quite a fundamental issue: looking at connectivity in a low dimensional space, that of the laplacian eigenvectors. There are two issues with this. The first one, which is less important than the second, is that the authors have a reference embedding to which they align other embeddings using a procrustes method with no scaling. While the 3D embedding is still optimally representing the connectivity (because distances don't change under rotations), we can no longer look at one axis at a time, which is what the authors do when they look at G1. In this case, G1 is representative of the connectivity of the reference matrix (LL), but not the others.But even if the authors only projected their matrices onto a single G1 dimension with no procrustes (and only sign flipping if necessary), there is still a major issue. One implicit assumption of this whole approach is that if there is a change in connectivity somewhere in the original matrix, the same "nodes" of the matrix will change in the embedding. This is not the case. Any change in the original matrix, even if it is a single edge, will affect the positions of all the nodes in the embedding. That is because the embedding optimises a global loss function, not a local one.To make this point clear, consider the following toy example. Say we have 4 brain regions A,B,C,D. Let us say that we have the following connectivity:In the Left Hemisphere: A-B-C-DIn the Right Hemisphere: A-B=C-DSo the connection between B and C is twice as strong in the right hemi, and everything else remains the same.The low dimensional embedding of both will look like this:Left:… A … B ……. C … D …RightA… … … B … C … … … DNote how B,C are closer to each other in the RIGHT, but also that A,D have moved away from each other because the eigenvector has to have norm 1.So if we were to calculate an asymmetry index, we would say that:A is higher on the LEFTB is higher on the RIGHTC is higher on the LEFTD is higher on the RIGHTSo we have found asymmetry in all of our regions. But in fact the only thing that has changed is the connection between B and C.This illustrates the danger of using a global optimisation procedure (like low-dim embedding) to analyse and interpret local changes. One has to be very careful.

We thank the Reviewer for the detailed description of the first concern. We agree that low-dimensional embeddings describe global embedding of local features, rather than local phenomena. Moreover, we indeed assume that the connectivity embedding of a given node gives us information about its position along ‘gradients’ relative to other nodes and their respective embedding. Thus, indeed, when a single node (node X) has a different connectivity profile in the right hemisphere relative to the left, this will also have some impact on the embeddings of all nodes showing a relevant (i.e., top 10%) connection to node X.

To evaluate whether asymmetry could be observed in average connectivity within functional networks, an alternative approach to measure asymmetry was taken by computing average connectivity within different functional networks. Following we compared the within-network connectivity between left and right. We have now added this conceptual analysis to our results robustness analysis section. In short, we observed that transmodal networks (DMN, FPN, and language network) showed higher connectivity in the left hemisphere but other networks showed higher connectivity in the right hemisphere. Thus, this indicates that observations made with respect to asymmetry of functional gradients are similar to those observed for within-network functional asymmetry between the left and right hemispheres. We have now detailed the outcome of this analysis in our Result section and Supplementary Materials.

Results, p.14.:

“As low-dimensional embedding is a global approach to summarize functional connectivity we reiterated our analysis by evaluating asymmetry of within network functional connectivity in the current sample. Observations made with respect to asymmetry of functional gradients are similar to those observed for within-network functional asymmetry between the left and right hemispheres.”

“To further explore functional connectivity asymmetry between left and right hemispheres, we calculated the LL within network FC and RR within network FC (Figure 2—figure supplement 5). It showed that connections in the left hemisphere and right hemisphere were relatively equal in the global scale. However, for the local differences, networks showed significant subtle leftward or rightward asymmetry (vis1: t = -5.203, P < 0.001; vis2: t = -22.593, P < 0.001; SMN: t = -8.262, P < 0.001; CON: t = -32.715, P < 0.001; DAN: t = -11.272, P < 0.001; Lan.: t = 33.827, P < 0.001; FPN: t = 24.439, P < 0.001; Aud.: t = 0.191, P = 0.849; DMN: t = 11.303, P < 0.001; PMN: t = -35.719, P < 0.001; VMN: t = -11.056, P < 0.001; OAN: t = 0.311, P = 0.756).”

Irrespectively, we have further highlighted that such a global interpretation for asymmetry of areas is still meaningful, given that a node is always placed in a global context. We have now further explained that our metrics give insights in local embedding of global phenomena in the introduction, p. 3.

Introduction, p. 3:

“These low-dimensional gradient embeddings describe global embedding of local features, rather than local phenomena. Thus, interpretation for asymmetry of areas is under a global context.”

My second concern is about interpreting the brain asymmetry as differences in connectivity, as opposed to differences in other things like regional size. The authors use a parcellated approach, where presumably the parcels are left-right symmetric. If one area is actually larger in one hemisphere than in the other, the will manifest itself in the connectivity values. To mitigate this, it may be necessary to align the two hemispheres to each other (maybe using spherical registration) using connectivity prior to applying the parcellation.

Thanks for this nice idea. We have now computed the differences of the mean rsfMRI connectome along the first gradient at the vertex level using 100 random subjects, as we have the data mapped to a symmetric template (fs_LR_32k), indicating that each vertex has a symmetric counterpart in the right hemisphere. Our results show left-right asymmetry as language/default mode-visual-frontoparietal vertices, which is consistent with the main results of the parcel-based approach. We have also added this response to the Supplementary materials.

Though overall findings are consistent, spherical registration may also have new issues. Total anatomical spatial symmetry may not provide functional comparability at the vertex level between left and right hemisphere. For example, during language tasks in the current sample, the activated frontal region in the left hemisphere is larger than the activated contralateral region in the right hemisphere. In the current study, we aimed to evaluate asymmetry between functionally and structurally homologous regions, as described by the Glasser atlas. In case of the resting state fMRI data, we used the region-wise symmetric multimodal parcellation (Glasser et al., 2016). This parcellation ensures the functional contralateral regions in both hemispheres. A previous study (Williams et al., 2021) investigated the structural and functional asymmetry in newborn infants. They used spherical registration (make fs_LR symmetric) for structural asymmetry but not for functional asymmetry. As such spheric registration may hide functional information, we think spherical registration may be more suitable for structural studies.

To address the concern regarding the alignment of hemispheres, we used joint alignment for LL and RR to compare the results between this and the Procrustes alignment technique (Pearson r=0.930, P_spin<0.001), Figure2—figure supplement 7 is the figure of asymmetry along the principal gradient (upper: joint alignment, below: Procrustes alignment) indicating convergence between both approaches. We have reported this information in the Supplementary Materials.

Lastly, we do agree that parcel size might be an important issue influencing the asymmetry pattern. To test for such an effect, we performed the correlation between the rank of parcel size (left-right)/(left+right) and rank of asymmetry index. It suggests only a small insignificant correlation along G1 (Spearman r_intra=0.130, P_spin=0.105; Spearman r_inter=0.130, P_spin=0.084). Of note, there is a systematic difference in parcel size as a function of sensory-association hierarchy, indicating that the link between parcel-size and asymmetry may vary as a function of sensory vs associative regions.

Figure 1. Please explain what "explained variance" means. The gradients represent a low dimensional version of the connectivity matrix. they are not explaining variance?

We thank the Reviewer for this question. The “explained variance” along the y-axis is the contribution of each eigenvector to the whole-brain connectivity variance decomposed but not the eigen-variance along (G1, G2, and G3 that are presented in the 3D scatter of Figure 1). We have revised the legend of Figure 1d about the “Variance %”.

Figure 1d, legend: “(d) Gradient template using the group-level gradient of LL. Dots represent parcels and were colored according to Cole-Anticevic networks. The decomposition scatter in the right below side depicts x-axis (number of eigenvectors) and y-axis (the contribution of each eigenvector to the total)”

Reviewer #2 (Recommendations for the authors):Using recently-developed functional gradient techniques, this study explored human brain hemispheric asymmetry. The functional gradient is a hot technique in recent years and has been applied to study brain asymmetries in two papers of 2021. Compared to previous studies, the current study further evaluated the degree of genetic control (heritability) and evolutionary conservation for such gradient asymmetries by using human twin data and monkey's fMRI data. These investigations are of value and do provide interesting data. However, it suffers from a lack of specific hypotheses/questions/motivations underlying all kinds of analyses, and the rich observational or correlational results seem not to offer significant improvement of theoretical understanding about brain asymmetries or functional gradient. In addition, given the limited number of twins in HCP project (for a heritability estimation), the limited number of monkeys (20 monkeys), and the relatively poor quality of monkeys' resting functional MRI data, the results and conclusion should be taken cautiously. Below are the concerns and suggestions.

We thank the Reviewer for the evaluation of our work and the helpful suggestions.

The gradient from resting-state functional connectome has been frequently used but mainly at the group level. The current study essentially applied the gradient comparison (i.e., gradient score) at the individual level. Biological interpretation for individual gradient score at the parcel level as well as its comparability between individuals and between hemispheres should be resolved. This is the fundamental rationale underlying the whole analyses.

We thank the Reviewer for this remark, and are happy to provide further rationale for using and comparing individual gradients scores to evaluate individual variation in asymmetry and associated heritability. Though gradients from resting-state functional connectivity have been frequently used at the group level, various studies have also studied individual differences. For example, using linear mixed models to compare gradient scores between left and right across subjects (Liang et al., 2021), applying the individual gradient scores to compare disease and controls (Dong et al., 2020, 2021; Hong et al., 2019; Park et al., 2021), and link individual hippocampal gradients to memory recollection (Przeździk et al., 2019). Together, these studies show individual variations of local gradients, indicating changes in node centrality and hubness (Hong et al., 2019), and connectivity profile distance (Y. Wang et al., 2021). Of note, low-dimensional embeddings describe global embedding of local features, rather than local phenomena. Thus, interpretation for asymmetry of areas is under a global context. The biological interpretation for individual gradients would be to what degree the system segregated and integrated has changed patterns of ongoing neural activity (Mckeown et al., 2020). It reflects that individuals have different functional boundaries between anatomical regions. Whereas individual neurons are embedded under the global-local boundaries through a cortical wiring space consisting of intricate long- and short-range white matter fibers (Paquola et al., 2020).

Introduction, p. 4:

“We applied the individual gradient scores to study the asymmetry, consistent with prior studies (Gonzalez Alam et al., 2021; Liang et al., 2021). Individual variation along the gradients reflects a global change across subjects in the functional connectome integration and segregation, and it is under genetic control (Valk et al., 2021). Moreover, to what degree the system segregated and integrated relates to patterns of ongoing neural activity (Mckeown et al., 2020), and different individuals have different functional boundaries between anatomical regions.”

Results, p. 5:

“Next, individual gradients were computed for each subject and the four different FC modes and aligned to the template gradients with Procrustes rotation. It rotates a matrix to maximum similarity with a target matrix minimizing sum of squared differences. As noted, Procrustes matching was applied without a scaling factor so that the reference template only matters for matching the order and direction of the gradients. Therefore, it allows comparison between individuals and hemispheres. The individual mean gradients showed high correlation with the group gradients LL (all Pearson r > 0.97, P spin < 0.001).”

Only the first three gradients are used but why? What about the fourth gradient? Specific theoretical interpretation is needed. At the individual level, is it ensured that the first gradients of all individuals correspond to each other? In this study, it is unclear whether we should or should not care about the G2 and G3. The results of G2 and G3 showed up randomly to some degree.

In the current study we focused on the principal gradient in the main analysis, given its association with sensory-transmodal hierarchy, microstructure, and evolutionary alterations (Margulies et al., 2016; Paquola et al., 2019; Xu et al., 2020).

Conversely, gradient 2 reflects the dissociation between visual and sensory-motor networks and gradient 3 is linked to task-positive, control, versus ‘default’ and sensory-motor regions. We analyzed asymmetry and its heritability of the first three gradients (explaining respectively 23.3%, 18.1%, and 15.0% of the variance of the rsFC matrix). However, we extracted the first ten gradients to maximize the degree of fit (Margulies et al., 2016; Mckeown et al., 2020). We have now also shown G4-10 mean asymmetry results as a supplementary figure. To ensure correspondence of gradients across individuals, we aligned the individual gradients to the group level template with Procrustes rotation. Procrustes rotation rotates a matrix to maximum similarity with a target matrix minimizing sum of squared differences. The approach is typically used in comparison of ordination results and is particularly useful in comparing alternative solutions in multidimensional scaling. Figure S1 shows the mean gradients across subjects of each FC mode, which is close to the Figure 1D template gradient space.

Results, p. 5:

“The current study analyzed asymmetry and its heritability of the first three gradients explaining most variance (Figure 1d). As they all have reasonably well described functional associations (G1: unimodal-transmodal gradient with 24.1%, G2: somatosensory-visual gradient with 18.4%, G3: multi-demand gradient with 15.1%). However, given we extracted ten gradients to maximize the degree of fit ^26,52^. We stated mean asymmetry of G4-10 in Figure 1—figure supplement 1.”

The intra-hemispheric gradient is institutive. However, it is hard to understand what the inter-hemispheric gradient means. From the data perspective, yes you can do such gradient comparison between the LR and RL connectome but what does this mean? Why should we care about such asymmetry? From the introduction to the discussion, the authors simply showed the data of inter-hemispheric gradients without useful explanation. This issue should be solved.

We are happy to further clarify. The LR and RL connectivity reflects cross-hemispheric functional signal interaction via corpus callosum, whose structural asymmetry is usually studied (Karolis et al., 2019). Such intra-hemispheric connections, compared to the inter-hemispheric connections, have been suggested to reflect the inhibition of corpus callosum, and underlie hemispheric specialization. Different information relies on hemispheric specialization (e.g., visual, motor, and crude information) and/or inter-hemispheric information transfer (e.g., language, reasoning, and attention) (Gazzaniga, 2000). To clarify and motivate the analysis of both intra- and inter-hemispheric asymmetry in functional gradients, we have now added further detail in the introduction, p. 5.

Here is text:

Introduction, p. 4.

“The full FC matrix contains both intra-hemispheric and inter-hemispheric connections. Intra-hemispheric connections, compared to the inter-hemispheric connections, have been suggested to reflect the inhibition of corpus callosum and may underlie hemispheric specializations involving language, reasoning, and attention. Conversely, inter-hemispheric connectivity may reflect information transfer between hemispheres, for example a wide range of modal and motor information, and crude information concerning spatial locations ^48^. Previous studies have reported intra-hemispheric FC to study gradient asymmetry ^6,38^. By having the callosum related to association white matter fibers, one hemisphere could develop for new functions while the other hemisphere could continue to perform the previous functions for both hemispheres ^48^. Therefore, in addition to the intra-hemispheric FC gradients, we depicted the inter-hemispheric FC, which is abnormal in patients with schizophrenia ^23,49^ and autism ^24^.”

as well as Discussion, p. 16

“Conversely, the transmodal frontoparietal network was located at the apex of rightward preference, possibly suggesting a right-ward lateralization of cortical regions associated with attention and control and ‘default’ internal cognition ^62,63^. The observed dissociation between language and control networks is also in line with previous work suggesting an inverse pattern of language and attention between hemispheres ^3,64^. Such patterns may be linked to inhibition of corpus callosum ^65^, promoting hemispheric specialization. It has been suggested that such inter-hemispheric connections set the stage for intra-hemispheric patterns related to association fibers ^48^. Future research may relate functional asymmetry directly to asymmetry in underlying structure to uncover how different white-matter tracts contribute to asymmetry of functional organization.”

and Discussion, p.18

“Though overall intra- and inter-hemispheric connectivity showed a strong spatial overlap in humans, we also observed marked differences between both metrics across our analysis. For example, although we found both intra- and inter-hemispheric differences in gradient organization to be heritable, only for intra-hemispheric asymmetry we found a correspondence between degree of asymmetry and degree of heritability. Similarly comparing asymmetry observed in human data to functional gradient asymmetry in macaques, we only observed spatial patterning of asymmetry was conserved for intra-hemispheric connections. Whereas intra-hemispheric asymmetry relates to association fibers, commissural fibers underlie inter-hemispheric connections ^77^ It has been suggested that there is a trade-off within and across mammals of inter- and intra-hemispheric connectivity patterns to conserve the balance between grey and white-matter ^76^. Consequently, differences in asymmetry of both ipsi- and contralateral functional connections may be reflective of adjustments in this balance within and across species. Secondly, previous research studying intra- and inter-hemispheric connectivity and associated asymmetry has indicated a developmental trajectory from inter- to intra-hemispheric organization of brain functional connectivity, varying from unimodal to transmodal areas ^78,79^. It is thus possible that a reduced correspondence of asymmetry and heritability in humans, as well as lack of spatial similarities between humans and macaques for inter-hemispheric connectivity may be due to the age of both samples (young adults in humans, adolescents in macaques). Further research may study inter- and intra-hemispheric asymmetry in functional organization as a function of development in both species to further disentangle heritability and cross-species conservation and adaptation.”

When aligning intra-hemispheric gradient, choosing averaged LL mode as the reference may introduce systematic bias towards left hemisphere. Such an issue also applies to LR-RL gradient alignment as well as cross-species gradient alignment. This methodological issue should be solved.

We thank the Reviewer for raising this point. Indeed, we also used RR as reference, the results were virtually identical. We have stated this in the Results, p. 13. Regarding the cross-species alignment, we averaged the left and right hemispheres to reduce the systematic bias. It showed that the correlation and comparison results remained robust. Now we have updated the method and corresponding results (p.10). Here is the text:

Results (p.15):

“We also set the RR FC gradients as reference, the first three of which explained 22.8%, 18.8%, and 15.9% of total variance. We aligned each individual to this reference. It suggested all results were virtually identical (Pearson r > 0.9, P spin < 0.001).”

Results (p.10):

“To reduce a possible systematic hemispheric bias during the cross-species alignment, we averaged the left and right hemisphere. We found that the macaque and macaque-aligned human AI maps of G1 were correlated positively for intra-hemispheric patterns (Pearson r = 0.345, P _spin_ = 0.030). For inter-hemispheric patterns, we didn’t observe a significant association (Pearson r = -0.029, P _spin_ = 0.858)”

The sample size of monkey (i.e., 20) is far less than human subjects (> 1000). Such limitation raises severe concern on the validity of the currently observed gradient asymmetry pattern in the monkey group, as well as the similarity results with human gradient asymmetry pattern. Despite the marginal significance of G1 inter-hemisphere gradient between humans and monkeys, I feel overall there is no convincingly meaningful similarity between these two species. However, the authors' discussion and conclusion are largely based on strong inter-species similarity in such asymmetry. The conclusion of evolutionary conservation for gradient asymmetry, therefore, is not well supported by the results.

We agree with your comments. Although it is a small sample compared to humans, in NHP studies, it is a relatively decent sample size (most of the studies have N<10). Of note, recent work suggested that the individual variation pattern can be captured using 4 subjects in both human and macaques (Ren et al., 2021).

To overcome potential overinterpretation of our findings, we have now changed the title to a more descriptive format:

“Heritability and cross-species comparisons of asymmetry of human cortical functional organization”

And further detailed findings already in the Abstract;

“These asymmetries were heritable in humans and, for intra-hemispheric asymmetry of functional connectivity, showed similar spatial distributions in humans and macaques, suggesting phylogenetic conservation.”

We have pointed out the small sample size in the limitation. Please find the text below:

Discussion, p. 18:

“Due to the small sample size of macaques, it is important to be careful when interpreting our observations regarding asymmetry in macaques, and its relation to asymmetry patterning observed in humans. Therefore, further study is needed to evaluate the asymmetry patterns in macaques using large datasets ^53,79^”

And nuanced the conclusion, p.19:

“This asymmetry was heritable and, in the case of organization of intra-hemispheric connectivity, showed spatial correspondence between humans and macaques. At the same time, functional asymmetry was more pronounced in language networks in humans relative to macaques, suggesting adaptation.”

For human gradient asymmetry, only t values were provided; For monkey gradient asymmetry, only Cohen-d values were provided. These two should be provided for both species.

Thanks for the comment. For the humans, we have now provided the Cohen’s d map as a supplementary figure.

Figure 3b, it is hard to believe that such a scatter plot can reach a significant correlation of R>0.3. In addition, such a scatter plot does not match the text (i.e., correlation between the "absolute" AI and heritability)

Thanks for pointing this out. In the previous version of the figure, we wanted to show whether a region showing relatively high heritability has a left or right-ward asymmetry, and computed their relationship using absolute values of asymmetry. We have now revised the corresponding figure, and added further detail in the results paragraph, please see below and Results Figure 3 section, p.9.

Results Figure 3, p. 9:

“To assess whether regions showing higher asymmetry had an increased heritability of G1, we plotted our cortical maps of asymmetry along those reporting heritability (Figure 3b). For the correlation between the absolute asymmetry index and heritability (Figure 3b small scatter), gradients of the intra-hemispheric FC patterns were significant (Pearson r = 0.245, P _spin_ = 0.005) but gradients of the inter-hemispheric FC were not (Pearson r = 0.055, P _spin_ = 0.613).”

Figure S3, why should we care about these cross-gradient correlations?

Thanks for your comment and we apologize when the motivation was not clear, originally we intended to transparently display whether there was any relationship between asymmetry G1-3 in humans and macaques. However, we agree with the Reviewer that this analysis is unclear, and moves away from the main question of the work. Thus, we have now removed this cross-gradient correlations figure and focused in particular on the G1 cross-species correlation.

More detailed description for fMRI post-processing for functional connectome and gradient analyses could be added in the supplementary information.

We are happy to further detail fMRI post-processing for functional connectome and gradient analysis, please find below. in the Methods;

FC, p.21:

“All rs-fMRI data underwent HCP’s minimal preprocessing ^80^ and were coregistered using a multimodal surface matching algorithm (MSMAll) ^83^ to the HCP template 32k_LR surface space. The template consists of 32,492 total vertices per hemisphere (59,412 excluding the medial wall). Cortical time series were averaged within a previously established multi-modal parcellation schemes: for humans the 360-parcel Glasser atlas (180 per hemisphere) ^54^ and the 182-parcel Markov atlas (91 per hemisphere) for macaques ^56^. To compute the functional connectivity (FC), time-series of cortical parcels were correlated pairwise using the Pearson product moment and then Fisher’s z-transformed in human and macaque data, separately. Individual FC maps were also averaged across four different rs-fMRI sessions for humans ([LR1], [LR2], [RL1], and [RL2]). We computed the FC in four different patterns, both for human and macaque data: FC within the left and right hemispheres (LL intra-hemisphere, RR intra-hemisphere), from the left to right hemisphere (LR inter-hemisphere) and from the right to left hemisphere (RL, inter-hemisphere).”

Connectivity gradients, p.21:

“Next we employed the nonlinear dimensionality reduction technique ^26^ to generate the group level gradients of the mean LL FC across individuals. We then set the group-level gradients as the template and aligned each individual gradient with Procrustes rotation to the template. Finally, the comparative individual functional gradients of each FC pattern were assessed. All steps were accomplished in the Python package Brainspace ^27^. In brief, the algorithm estimates a low-dimensional embedding from a high-dimensional affinity matrix. Along these low-dimensional axes, or gradients, cortical nodes that are strongly interconnected, by either many suprathreshold edges or few very strong edges, are closer together. Nodes with little connectivity similarly are farther apart. Regions having similar connectivity profiles are embedded together along the gradient axis. The name of this approach, which belongs to the family of graph Laplacians, is derived from the equivalence of the Euclidean distance between points in the diffusion embedded mapping ^25–27^. It is controlled by a single parameter α, which controls the influence of the density of sampling points on the manifold (α = 0, maximal influence; α = 1, no influence). On the basis of the previous work ^26^, we followed recommendations and set α = 0.5, a choice that retains the global relations between data points in the embedded space and has been suggested to be relatively robust to noise in the covariance matrix.”

“The input of the analysis was the FC matrix, which was cut off at 90% similar to previous studies ^26^. The current study selected the first three FC LL gradients (G1, G2, and G3) that explained 24.1%, 18.4%, and 15.1% of total variance in humans, as well as 18.9%, 15.2%, and 12.8% of total variance in macaques”

DK atlas is not a good validation parcellation for a functional MRI study like this.

We agree with the Reviewer. However, given that various structural MRI studies on brain asymmetry (e.g., Kong et al., 2018, 2022; Sha et al., 2021) have reported results using DK atlas, we have put this figure on the Figure 2 supplements for comparison of results and completeness with respect to this literature.

References

Dong, D., Luo, C., Guell, X., Wang, Y., He, H., Duan, M., Eickhoff, S. B., and Yao, D. (2020). Compression of Cerebellar Functional Gradients in Schizophrenia. *Schizophrenia Bulletin*, *46*(5), 1282–1295. https://doi.org/10.1093/schbul/sbaa016

Dong, D., Yao, D., Wang, Y., Hong, S.-J., Genon, S., Xin, F., Jung, K., He, H., Chang, X., Duan, M., Bernhardt, B. C., Margulies, D. S., Sepulcre, J., Eickhoff, S. B., and Luo, C. (2021). Compressed sensorimotor-to-transmodal hierarchical organization in schizophrenia. *Psychological Medicine*, 1–14. https://doi.org/10.1017/S0033291721002129

Gazzaniga, M. S. (2000). Cerebral specialization and interhemispheric communication: Does the corpus callosum enable the human condition? *Brain*, *123*(7), 1293–1326. https://doi.org/10.1093/brain/123.7.1293

Glasser, M. F., Coalson, T. S., Robinson, E. C., Hacker, C. D., Harwell, J., Yacoub, E., Ugurbil, K., Andersson, J., Beckmann, C. F., Jenkinson, M., Smith, S. M., and Van Essen, D. C. (2016). A multi-modal parcellation of human cerebral cortex. *Nature*, *536*(7615), 171–178. https://doi.org/10.1038/nature18933

Hong, S.-J., Vos de Wael, R., Bethlehem, R. A. I., Lariviere, S., Paquola, C., Valk, S. L., Milham, M. P., Di Martino, A., Margulies, D. S., Smallwood, J., and Bernhardt, B. C. (2019). Atypical functional connectome hierarchy in autism. *Nature Communications*, *10*(1), 1022. https://doi.org/10.1038/s41467-019-08944-1

Karolis, V. R., Corbetta, M., and Thiebaut de Schotten, M. (2019). The architecture of functional lateralisation and its relationship to callosal connectivity in the human brain. *Nature Communications*, *10*(1), 1417. https://doi.org/10.1038/s41467-019-09344-1

Kong, X.-Z. et al. Mapping cortical brain asymmetry in 17,141 healthy individuals worldwide via the ENIGMA Consortium. PNAS 115, E5154–E5163 (2018).

Kong, X.-Z. et al. Mapping brain asymmetry in health and disease through the ENIGMA consortium. Hum Brain Mapp 43, 167–181 (2022).

Liang, X., Zhao, C., Jin, X., Jiang, Y., Yang, L., Chen, Y., and Gong, G. (2021). Sex-related human brain asymmetry in hemispheric functional gradients. *NeuroImage*, *229*, 117761. https://doi.org/10.1016/j.neuroimage.2021.117761

Margulies, D. S., Ghosh, S. S., Goulas, A., Falkiewicz, M., Huntenburg, J. M., Langs, G., Bezgin, G., Eickhoff, S. B., Castellanos, F. X., Petrides, M., Jefferies, E., and Smallwood, J. (2016). Situating the default-mode network along a principal gradient of macroscale cortical organization. *Proceedings of the National Academy of Sciences*, *113*(44), 12574–12579. https://doi.org/10.1073/pnas.1608282113

Mckeown, B., Strawson, W. H., Wang, H.-T., Karapanagiotidis, T., Vos de Wael, R., Benkarim, O., Turnbull, A., Margulies, D., Jefferies, E., McCall, C., Bernhardt, B., and Smallwood, J. (2020). The relationship between individual variation in macroscale functional gradients and distinct aspects of ongoing thought. *NeuroImage*, *220*, 117072. https://doi.org/10.1016/j.neuroimage.2020.117072

Paquola, C., Wael, R. V. D., Wagstyl, K., Bethlehem, R. A. I., Hong, S.-J., Seidlitz, J., Bullmore, E. T., Evans, A. C., Misic, B., Margulies, D. S., Smallwood, J., and Bernhardt, B. C. (2019). Microstructural and functional gradients are increasingly dissociated in transmodal cortices. *PLOS Biology*, *17*(5), e3000284. https://doi.org/10.1371/journal.pbio.3000284

Park, B., Hong, S.-J., Valk, S. L., Paquola, C., Benkarim, O., Bethlehem, R. A. I., Di Martino, A., Milham, M. P., Gozzi, A., Yeo, B. T. T., Smallwood, J., and Bernhardt, B. C. (2021). Differences in subcortico-cortical interactions identified from connectome and microcircuit models in autism. *Nature Communications*, *12*(1), 2225. https://doi.org/10.1038/s41467-021-21732-0

Przeździk, I., Faber, M., Fernández, G., Beckmann, C. F., and Haak, K. V. (2019). The functional organisation of the hippocampus along its long axis is gradual and predicts recollection. *Cortex*, *119*, 324–335. https://doi.org/10.1016/j.cortex.2019.04.015

Ren, J., Xu, T., Wang, D., Li, M., Lin, Y., Schoeppe, F., Ramirez, J. S. B., Han, Y., Luan, G., Li, L., Liu, H., and Ahveninen, J. (2021). Individual Variability in Functional Organization of the Human and Monkey Auditory Cortex. *Cerebral Cortex*, *31*(5), 2450–2465. https://doi.org/10.1093/cercor/bhaa366

Sha, Z. et al. The genetic architecture of structural left–right asymmetry of the human brain. Nat Hum Behav 5, 1226–1239 (2021).

Wang, Y., Royer, J., Park, B., Wael, R. V. de, Larivière, S., Tavakol, S., Rodriguez-Cruces, R., Paquola, C., Hong, S.-J., Margulies, D. S., Smallwood, J., Valk, S. L., Evans, A. C., and Bernhardt, B. C. (2021). Long-range connections mirror and link microarchitectural and cognitive hierarchies in the human brain (p. 2021.10.25.465692). https://www.biorxiv.org/content/10.1101/2021.10.25.465692v1

Williams, L. Z. J., Fitzgibbon, S. P., Bozek, J., Winkler, A. M., Dimitrova, R., Poppe, T., Schuh, A., Makropoulos, A., Cupitt, J., O’Muircheartaigh, J., Duff, E. P., Cordero-Grande, L., Price, A. N., Hajnal, J. V., Rueckert, D., Smith, S. M., Edwards, A. D., and Robinson, E. C. (2021). Structural and functional asymmetry of the neonatal cerebral cortex (p. 2021.10.13.464206). bioRxiv. https://doi.org/10.1101/2021.10.13.464206

Xu, T., Nenning, K.-H., Schwartz, E., Hong, S.-J., Vogelstein, J. T., Goulas, A., Fair, D. A., Schroeder, C. E., Margulies, D. S., Smallwood, J., Milham, M. P., and Langs, G. (2020). Cross-species functional alignment reveals evolutionary hierarchy within the connectome. *NeuroImage*, *223*, 117346. https://doi.org/10.1016/j.neuroimage.2020.117346

[Editors' note: further revisions were suggested prior to acceptance, as described below.]

Reviewer #1 (Recommendations for the authors):Thanks for your replies to my comments, and sorry for the delay in getting this response to you.Regarding my first comment, i.e. interpretation of a change in position along the gradients: I am not sure I understand your reply.

We thank the Reviewer for these important concerns. Please find our revised response to the original question below.

Q1.1_Rv1: The first concern is to me quite a fundamental issue: looking at connectivity in a low dimensional space, that of the laplacian eigenvectors. There are two issues with this. The first one, which is less important than the second, is that the authors have a reference embedding to which they align other embeddings using a procrustes method with no scaling. While the 3D embedding is still optimally representing the connectivity (because distances don't change under rotations), we can no longer look at one axis at a time, which is what the authors do when they look at G1. In this case, G1 is representative of the connectivity of the reference matrix (LL), but not the others. But even if the authors only projected their matrices onto a single G1 dimension with no procrustes ( and only sign flipping if necessary) […]

We now project the matrices in a single G1 dimension without Procrustes alignment, and with flipped signs where needed, as suggested by the Reviewer. Comparing the gradient asymmetry with Procrustes alignment to the gradient without alignment resulted in virtually identical results for the HCP sample ( r _intra-hemisphere_ = 0.956 , r _inter-hemisphere_ = 0.843). At the same time, comparing unaligned and aligned gradients in the UK Biobank sample, we find that the alignment improves the similarity to the pattern observed in HCP (aligned r _intra-hemisphere_ = 0.592 , non-aligned r _intra-hemisphere_ = 0.487, aligned r _inter-hemisphere_ = 0.384, non-aligned r _inter-hemisphere_ = 0.162). We agree with the Reviewer that the Procrustes alignment procedure may create somewhat of a mixture between two vectors. However, we consider the impact of this operation minor and that in general Procrustus alignment comes with important benefits, namely to be able to make comparisons of multiple vectors across participants and samples. For example, using Procrustes alignment has been shown to have beneficial effects on the reproducibility and stability of gradients across techniques, parameters and individuals (Hong et al., 2020).

We now include an explicit comparison of Procrustes and non-aligned gradients in our supplemental materials, p. 15.

“To evaluate potential downstream effects of alignment to our results, we compared the gradient asymmetry with Procrustes alignment to the gradient without alignment. This resulted in virtually identical results for the HCP sample ( r _intra-hemisphere_ = 0.956 , r _inter-hemisphere_ = 0.843, Figure 2—figure supplement 7 ). At the same time, comparing unaligned and aligned gradients in the UKB sample, we found that the alignment improved the similarity to the pattern observed in HCP (aligned r _intra-hemisphere_ = 0.592, non-aligned r _intra-hemisphere_ = 0.487, aligned r _inter-hemisphere_ = 0.384, non-aligned r _inter-hemisphere_ = 0.162, Figure 2-figure supplement 8).”

*is higher on the LEFT*

*is higher on the RIGHT*

*is higher on the LEFT*

*is higher on the RIGHT*
So we have found asymmetry in all of our regions. But in fact the only thing that has changed is the connection between B and C.This illustrates the danger of using a global optimisation procedure (like low-dim embedding) to analyse and interpret local changes. One has to be very careful.

We thank the Reviewer for this extensive and thorough comment. To overcome potential differences in normalization of eigenvectors that may occur when computing LL_RR and LR_RL gradients separately we computed the gradient of LL_RR and LR_RL in the same model, under the assumption that if the homologous regions in the left and right hemisphere would have the same connectivity, they would have the same gradient loading. However, if their connectivity pattern is different, they would have a different loading yet differences would be normalized equally for LL_RR and LR_RL as they are part of the same model.

We find that, along the principal gradient, the observed normalized asymmetric map is highly similar to the non-normalized map used in the main analyses for intra-hemispheric (Pearson r = 0.956). Conversely, for the inter-hemispheric (Pearson r = 0.531) asymmetry patterns are still consistent, yet we find the similarity reduced. It is possible the difference between intra- and inter-hemispheric correspondence relates to more global differences in strength of connectivity comparing LR to RL FC, as reported also in (Raemaekers et al., 2018) resulting in more widespread differences between inter-hemispheric patterns of both embedding procedures.

We have now included this analysis in the supplementary results, p. 15 and as a supplementary Figure.

“Moreover, to overcome potential normalization biases associated with creating one gradient for each hemisphere, we performed an alternative analysis to create a gradient of the left and right hemisphere together. This assumes that regions with similar connectivity profiles have comparable loading in the gradient framework. Indeed, along the principal gradient, the observed normalized asymmetric map was highly similar to the non-normalized map used in the main analyses for the intra-hemispheric (Pearson r = 0.956) and inter-hemispheric (Pearson r = 0.531) asymmetry patterns (Figure 2-figure supplement 9). It is possible the difference between intra- and inter-hemispheric correspondence relates to more global differences in strength of connectivity comparing LR to RL FC, as reported also in the article 7 resulting in more widespread differences between inter-hemispheric patterns of both embedding procedures.”

To further display the underlying differences in connectivity profiles we furthermore projected the top 10% of connectivity patterns which were the basis for dimensionality reduction techniques used in the main analyses on the cortex. Thus, any change that may alter potential gradient loadings would fall within the top 10% of a regions’ connectivity profiles. We selected the asymmetric parcel (No.25: Peri-Sylvian language area) of and displayed their left-right difference in top 10% connectivity profiles between left and right hemisphere. We have now added this example to the supplementary figures, and refer to it in the results, p.8.

"The differences in gradient loadings (parcel No.25: Peri-Sylvian language area) reflect differences in connectivity profiles (top 10%) between LL versus RR, or LR versus RL respectively (Figure 2—figure supplement 1)."

You agreed that it is difficult to interpret these changes, given that they can represent changes occurring outside the region where the change is reported, but then the analysis you have done does not address this concern.

We thank the Reviewer for this comment, and hope that both our answers to previous questions (Q1.Rv1 and Q1.2_Rv1) have created further clarity. Please find further answers and clarifications below.

Instead, you calculated some other measure (which I am not sure what it is as it is not well described) and reported the asymmetry index using this new measure. If this new measure is more interpretable, then why do you need to use gradients? What information from the gradients is useful for the study of asymmetries? And how can we interpret changes in positions along the gradients? Simply saying that "interpretation for asymmetry of areas is under a global context" seems to me like sweeping the issue under the rug.

We thank the Reviewer for raising these important points. Indeed, we have now revisited this question and provided updated analyses which provide a more fitting answer to the raised concerns. Overall we believe that using a gradient framework to study asymmetry of functional brain organization has various benefits and provides us with novel insights into the asymmetric organization of functional connectivity that can be integrated within the wider literature of cortical organization. First of all, gradients provide a synoptic framework to capture smooth variations of connectivity patterns across the cortical mantle. Such differences have been linked to graph-theoretical markers such as degree centrality (Hong et al., 2019) , and dynamics (Park et al., 2021) as well as connectivity distance (Hong et al., 2019; Wang et al., 2022). Moreover, the principal gradient provides a coordinate framework spanning the cortex and reflecting the geodesic distance between primary and default regions, and relates to cortical microstructure and associated processing hierarchies (Huntenburg et al., 2018). In doing so, and in contrast to clustering or network-based approaches, the gradient framework provides a spatial ordering of functional brain networks, placing them along a gradual axis of connectivity variation reaching from sensory to transmodal areas. In the context of asymmetry of gradient loadings this would mean that a given region with a significant left-ward asymmetry along the first gradient (sensory-to-transmodal) has a connectivity profile more similar to the transmodal anchor than in the left hemisphere relative to the right. Consequently these regions are placed at different positions along the cortical hierarchy, providing novel insights in the system-level variations of the asymmetric brain. This data-driven framework enabled us to evaluate heritability, i.e. to what extent genetic factors impact variation across individuals in asymmetric organization of the cortex, and compare patterns between humans and macaques. Moreover, our framework can, in future studies, be used to evaluate overlaps between asymmetry of local cortical structure with the functional (and structural) asymmetric embedding of a given region.

To better understand the asymmetry between hemispheres, previous work has also employed the gradients framework (Gonzalez Alam et al., 2022; Liang et al., 2021) and found inter-hemispheric functional connectivity profile differences in frontoparietal and default mode networks. In the current work, we further optimized the approach by considering individual variation and using an atlas that considers contralateral homologous regions and is based on the current sample (Glasser, 2016). Indeed, contrary to a network level account of asymmetry, the gradient framework helps us to describe the asymmetry of a large-scale functional coordinate system spanning sensory to transmodal anchors using a data-driven approach. We have now further highlighted these arguments in the Introduction, p.3.

“Gradients provide a synoptic framework to capture smooth variations of connectivity patterns across the cortical mantle. They describe variations in genetic patterning ^33,3435^ , functional processes ^26,32,36^ , and are observed across species ^34,37,38.^ Gradients have been linked to graph-theoretical markers such as degree centrality ^39^ and microcircuit dynamics ^40^ as well as connec-tivity distance ^39,41.^ Moreover, the principal gradient describes the geodesic distance between primary and default regions, and relates to cortical microstructure and associated processing hierarchies ^42.^ In doing so, and in contrast to clustering or network-based approaches, the gradi-ent framework provides a spatial ordering of functional brain networks, placing them along a gradual axis of connectivity variation reaching from sensory to transmodal areas. In the context of asymmetry of gradient loadings this would mean that a given region with a significant left-ward asymmetry along the first gradient (sensory-to-transmodal) has a connectivity profile more similar to the transmodal anchor in the left hemisphere relative to the right. Consequently, these regions are placed at different positions along the cortical hierarchy, providing novel in-sights concerning the system-level variations in the asymmetric brain.”

Regarding the issue of using the Procrustes to the template and how that makes the gradients a worse representation of connectivity for the non-template matrices: I don't understand the reply here either. What is meant by joint alignment and how exactly does this address my concern?

We apologize for the confusion. Joint alignment is implemented based on spectral embedding. Here the embedding, rather than using the affinity matrices individually, is based on the joint affinity matrix (Vos de Wael et al., 2020). However, as we now have updated our reply to the original question to provide a more fitting answer we believe this analysis is no longer relevant and have again removed its outcome from the supplementary materials.

Reference

Gonzalez Alam, T. R. del J., Mckeown, B. L. A., Gao, Z., Bernhardt, B., Vos de Wael, R., Margulies, D. S., Smallwood, J., and Jefferies, E. (2022). A tale of two gradients: Differences between the left and right hemispheres predict semantic cognition. *Brain Structure and Function* , *227* (2), 631–654. https://doi.org/10.1007/s00429-021-02374-w

Hong, S.-J., Vos de Wael, R., Bethlehem, R. A. I., Lariviere, S., Paquola, C., Valk, S. L., Milham, M. P., Di Martino, A., Margulies, D. S., Smallwood, J., and Bernhardt, B. C. (2019). Atypical functional connectome hierarchy in autism. *Nature Communications*, *10* (1), 1022. https://doi.org/10.1038/s41467-019-08944-1

Hong, S.-J., Xu, T., Nikolaidis, A., Smallwood, J., Margulies, D. S., Bernhardt, B., Vogelstein, J., and Milham, M. P. (2020). Toward a connectivity gradient-based framework for reproducible biomarker discovery. *NeuroImage*, *223* , 117322. https://doi.org/10.1016/j.neuroimage.2020.117322

Huntenburg, J. M., Bazin, P.-L., and Margulies, D. S. (2018). Large-Scale Gradients in Human Cortical Organization. *Trends in Cognitive Sciences*, *22* (1), 21–31. https://doi.org/10.1016/j.tics.2017.11.002

Liang, X., Zhao, C., Jin, X., Jiang, Y., Yang, L., Chen, Y., and Gong, G. (2021). Sex-related human brain asymmetry in hemispheric functional gradients. *NeuroImage*, *229*, 117761. https://doi.org/10.1016/j.neuroimage.2021.117761

Park, B., Hong, S.-J., Valk, S. L., Paquola, C., Benkarim, O., Bethlehem, R. A. I., Di Martino, A., Milham, M. P., Gozzi, A., Yeo, B. T. T., Smallwood, J., and Bernhardt, B. C. (2021). Differences in subcortico-cortical interactions identified from connectome and microcircuit models in autism. *Nature Communications*, *12* (1), 2225. https://doi.org/10.1038/s41467-021-21732-0

Raemaekers, M., Schellekens, W., Petridou, N., and Ramsey, N. F. (2018). Knowing left from right: Asymmetric functional connectivity during resting state. *Brain Structure and Function*, *223* (4), 1909–1922. https://doi.org/10.1007/s00429-017-1604-y

Vos de Wael, R., Benkarim, O., Paquola, C., Lariviere, S., Royer, J., Tavakol, S., Xu, T., Hong, S.-J., Langs, G., Valk, S., Misic, B., Milham, M., Margulies, D., Smallwood, J., and Bernhardt, B. C. (2020). BrainSpace: A toolbox for the analysis of macroscale gradients in neuroimaging and connectomics datasets. *Communications Biology*, *3* (1), 1–10. https://doi.org/10.1038/s42003-020-0794-7

Wang, Y., Royer, J., Park, B., Vos de Wael, R., Larivière, S., Tavakol, S., Rodriguez-Cruces, R., Paquola, C., Hong, S.-J., Margulies, D. S., Smallwood, J., Valk, S. L., Evans, A. C., and Bernhardt, B. C. (2022). Long-range functional connections mirror and link microarchitectural and cognitive hierarchies in the human brain. *Cerebral Cortex*, bhac172. https://doi.org/10.1093/cercor/bhac172